# clembench: Using Game Play to Evaluate Chat-Optimized Language Models as Conversational Agents

**Kranti Chalamalasetti, Jana Götze, Sherzod Hakimov**
**Brielen Madureira, Philipp Sadler, David Schlangen[1] [*]**
Computational Linguistics, Department of Linguistics
University of Potsdam, Germany
[1]German Research Center for Artificial Intelligence (DFKI), Berlin, Germany
`first.last@uni-potsdam.de`

## Abstract

Recent work has proposed a methodology for the systematic evaluation of "Situated Language Understanding Agents"—agents that operate in rich linguistic and non-linguistic contexts—through testing them in carefully constructed interactive settings. Other recent work has argued that Large Language Models (LLMs), if suitably set up, can be understood as (simulators of) such agents. A connection suggests itself, which this paper explores: Can LLMs be evaluated meaningfully by exposing them to constrained game-like settings that are built to challenge specific capabilities? As a proof of concept, this paper investigates five interaction settings, showing that current chat-optimised LLMs are, to an extent, capable of following game-play instructions. Both this capability and the quality of the game play, measured by how well the objectives of the different games are met, follows the development cycle, with newer models generally performing better. The metrics even for the comparatively simple example games are far from being saturated, suggesting that the proposed instrument will remain to have diagnostic value. Our general framework for implementing and evaluating games with LLMs is available at https://github.com/clembench.

## 1 Introduction

There is an old joke:

*A guy has a dog that plays checkers. "My goodness," everyone says, "that's amazing. What a brilliant dog!" — "Not really," he replies, "I beat him four games out of five."*

This joke nicely reflects where we are with

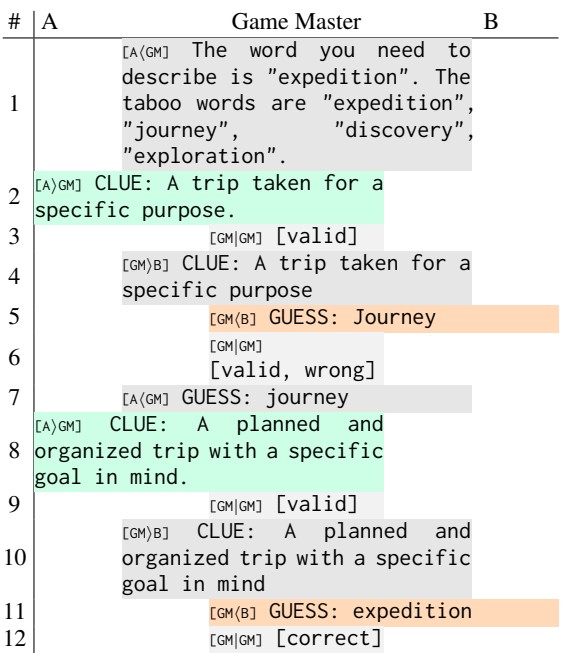

Figure 1: An episode of the `taboo` word game

interaction-tuned language models such as Chat-GPT and GPT-4 (OpenAI, 2023).[1,2] While the public discussion is dominated by what amounts to an unguided breadth-first search of tasks that can be "done" by these models (seeing "sparks" of generality in the process, Bubeck et al. (2023)), systematic investigations into how well these tasks are actually done, when looked at in depth, are only now beginning to appear (Liu et al., 2023; Bang et al., 2023)— often with results not dissimilar to what disappoints the dog owner in the joke, who apparently is looking for a challenging checkers partner and not a clever dog.

In this paper, we take the analogy even further and indeed look at how well these models can play interactive, language-based games, like that illustrated in Figure 1. In recent work, Schlangen

---

[*] Contributions: KC coded and analysed the `wordle` games. JG did so for `taboo`, and managed the code repository. SH coded and analysed `drawing` and `reference`, implemented the LLM interfaces, and edited the appendix. BM coded and analysed `private/shared`, and organised the evaluation computations. PS coded `taboo` and implemented the main framework. DS conceived the original idea, managed the project, and edited the main part of the paper.

[1]Thanks are due to Carl T. Bergstrom for bringing this joke and its applicability to the situation to our attention; https://fediscience.org/@ct_bergstrom/110273442253894015.

[2]We will call such models cLLMs from here on, for "chat-optimized LLM", with the suggested pronunciation "clem".

(2023a) has argued that such *Dialogue Games* ("constructed activities driven by language use") are a good systematic way of probing for the situated language understanding of language-using agents. In other recent work, Andreas (2022) has argued that LLMs are models of such agents. We bring these claims together and investigate what we can learn about the capabilities of cLLMs by exposing them to constrained game-like settings. Beyond making it possible to control the buildup of context in which to interpret the language, the game setting also has the advantage that we can generate novel instances that are unlikely to have been seen in any kind of training data, even if the game itself may have been. We describe a framework for implementing such games in a way that they can be tested in self-play of cLLMs—through the use of a programmatic "Game Master" that controls the game flow, as in the example in Figure 1—and we show results for five cooperative games that we have implemented in this framework, testing as game play agents the models Anthropic Claude, AlephAlpha Luminous, OpenAI GPT3, GPT3.5, GPT4 and open access ones such as Falcon, Open-Assistant, Vicuna and Koala.[3]

Our main findings are:

- Game instruction following in the best models generally is good, and is what marks the difference between models such as GPT-3 and newer models; likely as an effect of *instruction tuning* (Wei et al., 2022; Zhong et al., 2021) and learning from human feedback (Ouyang et al., 2022; Stiennon et al., 2020);
- The performance differences across games tracks the development cycle, with newer models generally performing better;
- The performance metrics are not saturated; and under the reasonable assumption that human performance would be near the ceiling, there is a wide gap between model performance and this.

Our contributions are:

- A flexible, extensible framework for the implementation of Dialogue Games as test instruments, which enables fast evaluation on a large (and extensible) set of models. The code repository is available via: https://github.com/

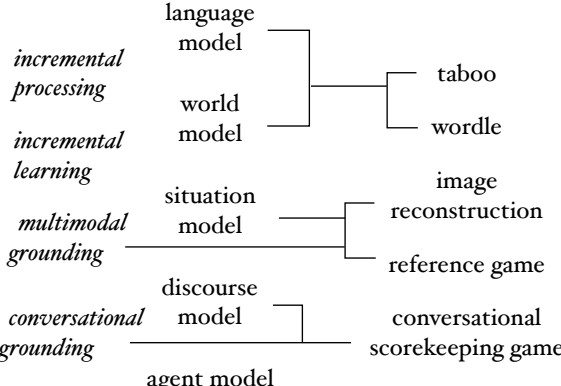

Figure 2: Anchoring Processes and Representational Domains from (Schlangen, 2023a,b) (left), and links to Dialogue Games described here

clembench.

- A collection of implemented and well-motivated games, together constituting version 1.0 of what we call the *clem benchmark*.
- An in-depth evaluation of the performance of current state-of-the-art cLLMs on these games.

## 2 Background: Situated Agents, Dialogue Games, and LLMs as Agent Models

Schlangen (2023a) introduces Dialogue Games as follows:

> A *Dialogue Game* is a constructed activity with a clear beginning and end, in which *players* attempt to reach a predetermined *goal state* primarily by means of producing and understanding linguistic material.

The claim is that such Dialogue Games can serve as valid instruments for evaluating models of situated language understanding, provided that an argument can be given for how a specific game challenges aspects of the underlying construct. As a model of this (not directly observable, but to be measured) construct he proposes what is illustrated here in in Figure 2, which analyses *situated language understanding* into a number of representational and procedural demands. Rather than going through these in detail here, we will illustrate them through the discussion of how the implemented games challenge these various aspects.

Andreas (2022) argues that LLMs "infer approximate, partial representations of the *beliefs, desires,* and *intentions* possessed by the agent that produced the context". If that is so, and if the finer-grained

---

[3]gpt4: (OpenAI, 2023); gpt3.5: (Ouyang et al., 2022); gpt3: (Brown et al., 2020); claude: (Bai et al., 2022); luminous-supreme: (AlephAlpha, 2023); falcon-40b-instruct (Almazrouei et al., 2023), open-assistant-12b (Köpf et al., 2023), vicuna-12b (Chiang et al., 2023) and koala-13b (Geng et al., 2023)

analysis of the relevant beliefs, desires, and intentions involved in game play that we reviewed in the previous paragraph is on the right track, then such games should form a valid instrument for measuring the degree to which LLMs do indeed approximate these capabilities.

Figure 2 illustrates how the example games implemented and evaluated here connect to the construct. (All games require a minimal form of discourse model being built, insofar as earlier information constrains later moves; and all games require a minimal type of agent model, insofar as the game instructions need to be taken on as own "intentions".) We will argue for these connections in detail below, but first we need to describe the scaffolding required to turn LLMs into game players.

## 3 From Game To Benchmark

### 3.1 Terminology

First, some terminology: A *Dialogue Game Realisation* (DGR) fixes for a given game the prompt templates (with which the game is described to the players) and the logic of the *Game Master* (the programmatic component keeping the game on track; see below). An *instance* of a DGR fixes the goal (e.g., in a word-guessing game, the word to guess) and the configuration. A *data set* is a collection of instances. An *experiment* fixes the players that get to play through a data set; e.g., as either being a human participant, or as a computer model (with all its free parameters fixed). For each *episode* (play of an instance), the experiment results in an *interaction record*. This record is what gets evaluated, both at a turn-by-turn level (progress made in the game) as well as for whether (or to what degree) the goal was reached. The *benchmark* then is a specific collection of datasets, and a *benchmark result* is the evaluation of (the interaction records of) a fixed combination of players over the benchmark.

### 3.2 Turn-Based Text Games via Prompting

Not all kinds of Dialogue Games in the sense of Schlangen (2023a) can be realised with LLMs as players. For now, the games need to be text-based (although we do realise games below that use character-encodings for image-like structures), and they need to be turn-based, so that each turn can be one prompting of a player to produce its move for this turn. We realise single-player games as well as two-player games. In order to keep the interaction focussed on the interactional task / the

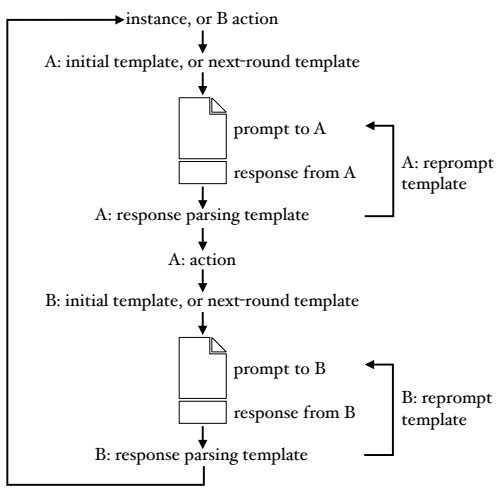

Figure 3: Schematic View of the Game Flow

Dialogue Game, we insert a (programmatic) *Game Master* into the interaction, whose task it is to keep track of the game state and to parse the reactions by the player, ensuring that only game-relevant actions are passed on and that the rules of the game are followed. In the taboo game shown in Figure 1 for example, the Game Master checks that the description given by player A does indeed not contain the "taboo" words (see description of the game below), before passing the message on to player B. In general, the Game Master is responsible for parsing the responses of the players and ensuring their formal adequacy. (At the moment, this is rather strict, leading to even slight variations being disregarded. At this point, we see this as still preferable over requiring more content-based parsing.) Thereby, the "general purpose" nature of any participating model is hidden, and it can be evaluated purely for its performance in the game.

The games considered here are self-contained in the sense that each game opens with a description of the game rules and instructions regarding the *form* of the response expected from a player; the game play consists in the player choosing the *content* of the move. This makes it possible to separately evaluate the ability to play the game (follow the instructions) and the level of expertise at playing it (e.g., by how fast or how well the goal has been reached in a given episode). Figure 3 shows a schematic view of how the Game Master controls a two-player game by making use of prompt templates that are filled in based on the current game state.

### 3.3 The `clemgame` Framework

We have implemented a Python framework that provides the general pattern (prompting, Game Master) described above, and takes care of the infrastructure of routing the player turns to the various model APIs (or, in the case of human players, to an appropriate interface). It is easily extensible to include new language-processing models (of type "string to string"; that is, models that can be prompted with a context and that return text). The framework also takes care of the separation of instance collections into datasets, of running (with different model settings) the experiments constituting the benchmark and evaluation based on scoring. All games described in the next section are implemented in this framework.

### 4 The Games in v1.0 of the Benchmark

All games described here challenge the rule-following capabilities of the players. In all games, the game objectives and the rules, including formal constraints on the game moves, are described verbally to the player. What these instructions leave implicit are general strategic considerations of game play, such as that repetitions in a guessing game have no strategic value. The Game Master validates each player move according to the formal constraints, and if after a certain amount of reprompting still no valid move is produced, the game is aborted. We measure for all games the proportion of games that were aborted in this way, giving us for each player a measure of their general ability to follow rules.

In the following, we briefly describe each game in general terms and define for each game a *quality score* with which to quantify the players' level of competence of playing it (beyond just following the rules so as to avoid the game play being aborted). Note that these metrics typically evaluate the pair of players together and cannot make role-based distinctions. All further details, such as how we realised the game through prompts and how we instantiated the realisation into game instances, are collected in the Appendix. Note that we did not specifically optimise the prompts for performance of any given model; we just made sure that our reference model GPT-3.5 seemed to be able to follow them. In any case, all models are challenged with exactly the same prompts for each game instance, ensuring validity of the relative outcomes. Other metrics common to all games are described in Appendix B. The games described here are those we selected for a first version of the benchmark, with the aim of breadth with respect to the model in Figure 2; we see as an advantage of the framework that it is easy to implement more games, and expect the benchmark to be extended through contributions from the community.

### 4.1 A Simple Word Game: Taboo

In this game, one player has to describe to another player a concept, without using the concept name and any of a given list of semantically related words. The task of the other player then is to guess this word. If the player guesses wrongly, the first player can attempt a different description, following the same constraints. To play this game, the players must be able to access information from the interface between what we called above the *language model* and the *world model*, to provide and understand concept descriptions. The additional constraints on word use challenge a player's ability to apply given rules to the given situation.

We have seen an example of game play already above in Figure 1, which indicated that the role of the Game Master is to provide the target word to player A, and to validate the players' moves (A cannot use taboo words; B either guesses correctly or not). We allow three rounds of guessing. The *quality score* for taboo is *speed*, scored non-linearly as $\frac{100}{n}$ (with $n$ as the number of required moves), or 0 (if $n > 3$). For Figure 1 hence this would yield 50.

### 4.2 Word-Guessing w/ Letter-Based Feedback

We also implemented some variations of the popular word-guessing game "Wordle".[4] The basic mechanics of this game is that letter-based feedback is provided on guesses of 5-letter words, which incrementally constrains the set of possible words. If the target word for example is APPLE, a guess of ALONE would yield the following information: A appears at this position, L appears elsewhere, O does not occur, N does not occur, E occurs at this position. We also implement non-standard variations where a textual clue is given at the beginning as to the identity of the word. These games are one-player games (although we technically realised the computation of letter-feedback as the contribution of a player B). We also implemented a variant where there is a more active additional player, who can give feedback on the choice of player A be-

---

[4]https://www.nytimes.com/games/wordle

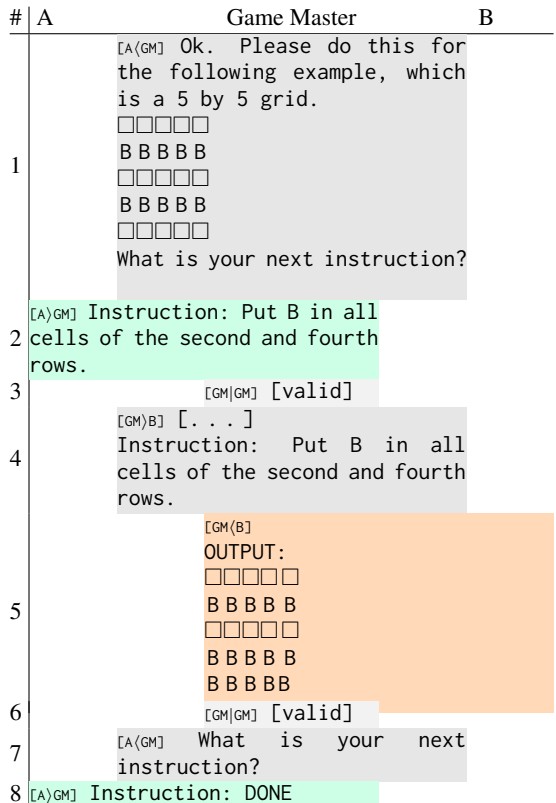

Figure 4: Excerpt of *wordle+clue+critic* game play (GPT4/GPT4)

fore it is played, giving A the opportunity to select differently. These game variants again challenge knowledge from *language* and *world model*, as well as, in a rudimentary form, in the "critic" variant, simulating processes of conversational grounding / negotiation. Figure 4 shows an excerpt of a game played with critic. The *quality score* for all variants again is *speed* (with a maximum of 6 guesses).

### 4.3 Drawing Instruction Giving and Following

In this game, player A is given an image (here represented as a grid of characters, with □ representing an empty cell), and their task is to instruct player B to reconstruct this image, starting from an empty grid. (See Figure 5 for an example.) Hence, to be successful both player A and B must form, in a limited multimodal way, a model of a (very much abstracted) situation. The game stops when player A signals that their description is complete. The *quality score* is the *F1-score* of player B's grid relative to player A's target and the non-empty "pixels": if all, and only, the target cells have been changed as desired, it is 100, if none have, it is 0. We test with *compact* instances, which allow for higher level descriptions (as in the example), and *random*

grids, which do not; see Appendix E.

Figure 5: An episode of the drawing game

### 4.4 A Picture Reference Game

We also implemented a simple *Lewis Signalling Game* (Lewis, 1969), where A is presented with three grids (of the type also used in drawing; shown in Figure 6) and the task to make B (who is also presented with the same three grids, but potentially in a different order) identify a pre-specified one. As in drawing, this game challenges the formation of a *situation model*, and, to be done efficiently, needs access to analogical information from the agent's *world model* (e.g., to describe the second grid in Figure 6 as "looks like a T"). There is a long tradition in psychology to use such reference games to provide insights into communicative behaviour (see, e.g., (Yule, 1997)). The *quality score* for this game is a simple binary *success* measure: Did B identify the target, or not?



Figure 6: Sample grids for the reference game

## 4.5 Scorekeeping: Private and Shared

| # | A | Game Master | B |
|---|---|---|---|
| 1 | | [A⟨GM] [initial prompt] | |
| 2 | | [GM⟨B] What class do you prefer? | |
| 3 | [A⟩GM] ANSWER: Economy. | | |
| 4 | | [GM⟨B] Which means of transportation do you prefer? | |
| 5 | [A⟩GM] ANSWER: Train. | | |
| 6 | | [GM⟨B] Please inform your destination. | |
| 7 | [A⟩GM] ANSWER: Stuttgart. | | |
| 8 | | [GM⟨B] What is the origin of your trip? | |
| 9 | [A⟩GM] ANSWER: London. | | |
| 10 | | [GM⟨B] When is the trip? | |
| 11 | [A⟩GM] ANSWER: In May. | | |

Figure 7: An example of the primary interaction in `private/shared`

The final game in v1.0 is structurally a bit different, as besides the main dialogical interaction, there is a secondary *probing interaction* going on, which is the target of interest here. In the primary interaction, an *answerer agent* goes through a form with a *questioner*. In the probing interaction, the game master probes the answerer agent on what they "think" that the questioner agent already knows. Each instance specifies the list of information that is to be exchanged, and so the game master can keep track of which information has already been shared. If a piece of information has not yet been shared, the answerer should not expect the questioner to know it. This game challenges the keeping of an *agent model* and its update through *conversational grounding* processes. Figure 7 shows an edited transcript of the primary interaction, and Figure 8 shows an example of a probing interaction that the game master can interject. For evaluation, we compute the slot-filling accuracy throughout the main interaction and the agreement between the model's answers and the ground truth in the probing rounds. Because each probe is a binary decision (shared or not), the random performance would be high, so we use Cohen's $\kappa$ (Cohen, 1960) to control for chance. The *quality score* is the harmonic mean between the slot-filling accuracy and the probing $\kappa$ (truncated at 0).

## 5 Results

As detailed in the Appendix, the full benchmark (v1.0) consists of 250 instances: 30 for `taboo`, 30 for `wordle`, 30 for `wordle+clue`, 30 for `wordle+clue+critic`, 40 for `drawing`, 40 for `reference`, and 50 for `private/shared`.

| # | A | Game Master | B |
|---|---|---|---|
| 1 | | [A⟨GM] ME: Do you think the travel agent knows where you depart from? Please answer yes or no. | |
| 2 | [A⟩GM] ASIDE: No. | | |
| 3 | | [A⟨GM] ME: Do you believe that the travel agent is aware of where you are going? Please answer yes or no. | |
| 4 | [A⟩GM] ASIDE: Yes. | | |
| 5 | | [A⟨GM] ME: Does the travel agent know how you wish to travel? Please answer yes or no. | |
| 6 | [A⟩GM] ASIDE: No. | | |
| 7 | | [A⟨GM] ME: Have you informed the travel agent about your class preference? Please answer yes or no. | |
| 8 | [A⟩GM] ASIDE: Yes. | | |
| 9 | | [A⟨GM] ME: Is the travel agent aware of the dates of your trip? Please answer yes or no. | |
| 10 | [A⟩GM] ASIDE: No. | | |

Figure 8: An example of the secondary interaction in `private/shared`; the model sees each question separately, with the primary dialogue as context

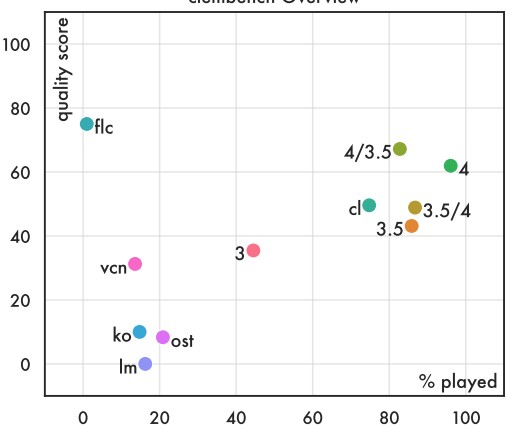

Figure 9: Overview of benchmark results

| model | version | here | P | T | I |
|---|---|---|---|---|---|
| gpt-4 | 0314 | 4 | n/a | n/a | Y |
| gpt-3.5-turbo | 0301 | 3.5 | n/a | n/a | Y |
| text-davinci | 003 | 3 | 175 | 300 | Y |
| claude | v1.3 | cl | 52 | n/a | Y |
| luminous-supreme | 2023-01 | lm | 70 | 588 | Y |
| falcon-40b-instruct | 2023-06 | flc | 40 | 600 | Y |
| vicuna-13b | 2023-06 | vcn | 13 | 1.4k | Y |
| open-assistant-12b | 2023-06 | ost | 12 | 400 | Y |
| koala-13b | 2023-06 | ko | 13 | 1.4k | Y |

Table 1: The evaluated models with the details about number of parameters in billions (P), trained data size (tokens) in billions (T), and whether they were instruction tuned (I). Y: yes, n/a: publicly not available.

We ran the benchmark on the models (closed and open-access) shown in Table 1 with self-play (a model plays all non-static players in a game). In

| | | all | taboo | wordle | wordle+cl | wordle+cr | drawing | reference | priv/sh |
|---|---|---|---|---|---|---|---|---|---|
| **lm/lm** | % played | 16.24 | 0.0 | **100.0** | 3.33 | 10.34 | 0.0 | 0.0 | 0.0 |
| 0.00 | qlty score | 00.00 | / | 0.0 (0.0) | 0.0 (-) | 0.0 (0.0) | / | / | / |
| **ko/ko** | % played | 14.76 | 0.0 | 86.67 | 16.67 | 0.0 | 0.0 | 0.0 | 0.0 |
| 1.47 | qlty score | 10.00 | / | 0.0 (0.0) | 20.0 (44.72) | / | / | / | / |
| **flc/flc** | % played | 0.95 | 0.0 | 0.0 | 3.33 | 3.33 | 0.0 | 0.0 | 0.0 |
| 0.71 | qlty score | 75.00 | / | / | **50.0** (-) | **100.0** (-) | / | / | / |
| **ost/ost** | % played | 20.85 | 0.0 | **100.0** | 16.67 | 14.29 | 0.0 | 15.0 | 0.0 |
| 1.73 | qlty score | 8.33 | / | 0.0 (0.0) | 0.0 (0.0) | 0.0 (0.0) | / | 33.33 (51.64) | / |
| **vcn/vcn** | % played | 13.58 | 5.08 | 56.67 | 13.33 | 20.0 | 0.0 | 0.0 | 0.0 |
| 4.24 | qlty score | 31.25 | **100.0** (0.0) | 0.0 (0.0) | 25.0 (50.0) | 0.0 (0.0) | / | / | / |
| **cl/cl** | % played | 74.76 | 76.92 | **100.0** | **100.0** | 46.43 | 0.0 | **100.0** | **100.0** |
| 37.06 | qlty score | 49.58 | 68.75 (38.71) | 0.0 (0.0) | 30.56 (40.13) | 30.77 (48.04) | / | **82.5** (38.48) | 84.87 (18.87) |
| **3/3** | % played | 44.50 | 28.81 | 66.67 | 36.67 | 23.33 | 57.5 | 82.5 | 16.0 |
| 15.77 | qlty score | 35.46 | 76.47 (43.72) | 1.25 (5.59) | 31.36 (38.99) | 50.0 (50.0) | 38.7 (27.78) | 36.36 (48.85) | 14.1 (25.21) |
| **3.5/3.5** | % played | 85.86 | 69.49 | **100.0** | 93.33 | 76.67 | **97.5** | **100.0** | 64.0 |
| 37.02 | qlty score | 43.12 | 71.95 (44.79) | 0.0 (0.0) | 28.57 (46.0) | 13.19 (30.16) | 60.28 (25.95) | 55.0 (50.38) | 72.83 (13.07) |
| **3.5/4** | % played | 86.75 | 69.49 | (single pl.) | (single pl.) | 80.0 | **97.5** | **100.0** | / |
| 42.39 | qlty score | 48.87 | 62.6 (45.15) | / | / | 10.42 (17.42) | 64.95 (25.45) | 57.5 (50.06) | / |
| **4/3.5** | % played | 82.78 | 66.1 | (single pl.) | (single pl.) | **100.0** | 65.0 | **100.0** | / |
| 55.61 | qlty score | **67.19** | 93.59 (23.45) | / | / | 46.67 (42.92) | 81.0 (21.54) | 47.5 (50.57) | / |
| **4/4** | % played | **96.06** | **94.92** | **100.0** | **100.0** | **100.0** | 77.5 | **100.0** | **100.0** |
| **59.48** | qlty score | 61.93 | 76.19 (37.45) | **3.67** (8.4) | 49.67 (42.09) | 49.11 (38.46) | **89.06** (22.28) | 75.0 (43.85) | **90.79** (8.2) |

Table 2: Results Overview. For each model (pairing), shows how many games were played to completion (% played), an indicator of rule-following capabilities. "qlty score" indicates how well the completed games were played (higher is better, max is 100; standard deviation in parentheses). all is the average over all games scores, the remaining columns show results broken down by game (averaged over all episodes). Values below model names are their clemscore. Updates / additional models posted at https://github.com/clembench.

addition, we run pairs of gpt-4 and gpt-3.5 to test if a supposedly better model (here gpt-4) can leverage the other. Following Srivastava et al. (2022), we requested greedy sampling (i.e., temperature 0). One run of the benchmark, somewhat surprisingly, on average took more than 600 minutes to complete, due to API latency, and cost around 50$ in API fees.

Table 2 gives the overall results of this benchmark run. For each model (pairing), we first show what percentage of instances were played to completion (i.e., not aborted because of problems of the players in following the instructions). We then show what the average quality of the play was for those played instances, using each game's quality score. The first column (macro-)averages the numbers over all games, with the remaining ones giving the per-game results. Figure 9 provides the same information in a graphical format, plotting "percentage played" against "quality". A perfect model—and, we suspect, since these are simple games, human performance—would be clustered in the top right corner (all instances played, with high quality). As we can see from the results, the GPT family tends to perform better than the other models we tested, with an increase in quality from 3 to 3.5 to 4. There is a jump in the ability to play games to completion (that is, to follow the prompt instructions as to the *format* of the game play moves) from 3 to 3.5, with a smaller increase from 3.5 to 4. Still, even the best performing model, GPT-4, does not reach 100% on "percentage played", with the reduction mostly due to drawing and, somewhat surprisingly, taboo — perhaps due to the *negative* nature of the game constraints ("don't say X").

When it comes to the quality of the game play (in those episodes played to completion), we see a similar trend, with GPT4 overall performing best. We also see that there is ample room for improvement, with the best average score standing at 60.59. An outlier in terms of quality is wordle, where even though almost all models manage to stick to the local rules (produce a 5-letter word), even the best-performing model, GPT4, only reaches

4.56 on the quality metric, indicating that very few games are actually solved, and those only at the last attempt. This indicates that all models fail at integrating the feedback across turns and using it to constrain their guesses. The capabilities of dealing with verbal meaning definitions are shown by the large improvement that wordle+clue exhibits (to 47.89). Interestingly, GPT4 is able to profit from (via another instance, self-)criticism, improving further to 50.11.

Again somewhat surprisingly, performance on the "multimodal" games (which require verbalisation of character-based graphics) is not bad. For drawing, as a deviation from the trend, GPT3.5 proved to be better at sticking to the game format (97.5% of episodes played to completion), although GPT4 still reached higher quality on its completed games. reference sees Claude performing best, against the trend for all other games.

## 6  Discussion

Insights from Taboo game: The results show that the open source models and Luminous cannot, or only badly, play the taboo game. Claude shows a strong performance with 76.92% played games and a quality score of 68.75%. The best scores are achieved by the GPT-4/4 pair with 94.92% played games and a quality score of 76.19%. We hypothesise that this is an effect of RLHF training time so that the model is highly aligned with the prompts given by the user (the game master). An indication is given by the increase from 28.81% to 69.49 % of games played when comparing GPT-3 (a foundation model) and GPT-3.5 (an RLHF fine-tuned model) where the quality score remains similar. Both models share the knowledge representation and are similarly good in retrieving that knowledge, but the latter is better aligned.

Insights from Reference and Drawing games: Claude and GPT 3.5 and 4 models get the best *played ratio*, which indicates that the generated outputs match the expected format. As this game is single turn, unlike the other games, errors cannot accummulate over turns. In *Drawing*, Luminous, Claude and the open access models did not manage to follow instructions. The generated outputs included the repetition of the text in given instructions, which leads for games to be aborted. The played ratio of GPT 3.5 is higher than GPT-4. By looking at some selected instances, we saw that the outputs from GPT-4 are simply the appended

text of multiple turns (on Player A side) instead of generating each instruction separately in a single turn. GPT-3.5 is better at following the instructions (97.5 vs. 77.5 in played score) but GPT-4 is better at getting the target drawing (60.2 vs 89.0 in quality score), in those cases where the format was correct.

Insights from the Scorekeeping game: Games were aborted mostly due to the models failing to use the correct player tag. This is particularly complicated in this game because we are trying to simulate a multi-party conversation, with one tag for each interlocutor. Interestingly, sometimes a mere reprompt with a generic addition (*e.g.* "Please answer this question carefully.") would trigger it to generate the right tag, even though the mistake was not added to the history. Another issue is that sometimes models would anticipate or invent slots and upcoming turns. Anticipating is not totally incorrect, but it makes it harder for the GM to check for the private/shared status of a slot. Claude and GPT-4 played the slot filling part very well; their mistakes came mostly from the scorekeeping component, with mixed results in abstract and concrete domains. In almost all cases, their main type of mistake was considering shared slot values to be still private.

Insights from the Wordle game: Models other than GPT-4 could not adhere to the game rules in at least half of the episodes they played. A significant observation is that most of these models did not incorporate the letter feedback to enhance their subsequent word guesses. This is evident from the repetition of letters from previous guesses in subsequent iterations. Figure 19a illustrates this observation. The turn at which a correct guess is made provide insights into the efficiency of the guessing strategy. In the traditional Wordle variant, GPT-4 takes an average of four turns (refer to Table 4 speed metric) to guess correctly, while it improves to two turns in extended variants. The presence of clue and feedback from the critic both improve the success rate and speed for the GPT-4 model. On the other hand, for other models the *Played* score degrades in the extended variants.

## 7  Related Work

Playing games and learning from self-play stands at the beginnings of the "deep learning revolution" (Mnih et al., 2013; Silver et al., 2017).[5]  What is

---

[5]Just as playing games stands at the beginning of machine learning in general, (Samuel, 1959).

different here is the zero- or few-shot nature of our test, where the testing mode is different from the learning mode—this of course only being enabled by "foundation models" (Brown et al., 2020). The latest—apparent—qualitative jump has only recently been taken, so there are not that many papers yet that attempt a systematic evaluation; see, *inter alia*, (Liu et al., 2023; Bang et al., 2023). To our knowledge, game play of the kind proposed here has not yet been used for the systematic evaluation of these models. The idea of testing game play is mentioned in (Bang et al., 2023; Bubeck et al., 2023), and also already technically possible in (Srivastava et al., 2022), but has not been systematically executed there.

A superficial similarity also exists to approaches like HuggingGPT (Shen et al., 2023) in that these approaches pair LLMs with scaffolding (as in our Game Master). A crucial difference, however, is that for us the task of the Game Master is to *constrain* the LLM and to "keep it focused", as it were, on the game, rather than to *extend* its capabilities.

Park et al. (2023) also acted on the realisation that cLLMs can simulate agents which can be put into "self-play", but developed this idea in a different direction, towards investigating the emerging "social behaviour".

Newly proposed benchmarks such as AlpacaEval (Li et al., 2023), Chatbot Arena (LMSYS, 2023) and Open LLM Leaderboard (HuggingFace, 2023) focus on comparing models outputs either running them on existing datasets, employ human annotators to choose which output is preferred, or simply ask another LLM to evaluate the outputs; these benchmarks do not test the interactive dialogue aspects of chat-based LLMs. Another important aspect to note here is that using existing datasets for benchmarking might jeopardise the point of keeping the test instances unseen because those instances could have been part of the training data for these large language models. The datasets for *clembench* have been created from scratch and adding new games or new instances to the existing games is easy to ensure continued fair benchmarking.

## 8  Roadmap

Important next steps on our roadmap include testing the models' abilities to handle languages other than English and integrating the framework with the slurk chat tool (Götze et al., 2022) in order to enable game play with human players. We also plan to experiment with games that have more than two players as well as games that require multimodal context such as images. We are also excited about the potential to use this as an instrument for testing models across size variations and training checkpoints, to analyse what it takes to acquire the capabilities tested here. Lastly, with the measuring instrument introduced here in place, we can also turn to improving individual models (rather than testing existing models out of the box) so as to optimise their performance on a particular game or set of games.

## 9  Conclusions

We have shown that current chat-optimised large language models can indeed serve as models of interactive agents, at least for controlled and rule-constituted activities such as verbal games. We have described our general implementation of a framework for implementing rules to be played in "self-play" by such models, with the main idea being that a programmatic component, the "Game Master" can control the interaction and ensure that only formally correct moves are registered. We have described our example implementations and instantiations of such games, arguing that they span the breadth of the sub-capabilities involved in situated language processing (if only on a relatively superficial level). Finally, we have shown that the evaluation of the game play can serve as an instrument to distinguish between models in terms of their language capabilities. With this work, we have aimed to open a complementary avenue for evaluating these models, beyond more classical reference-based NLP task evaluation or preference-based evaluation, and into the realm of interactive language use. Much remains to be done, but we hope that our framework can support some of this future work.

## Acknowledgements

The work reported here has been partially funded by the *Deutsche Forschungsgemeinschaft* (DFG, German Research Foundation), grants 423217434 ("RECOLAGE") and 317633480 (SFB 1287); and by *Bundesministerium für Bildung und Forschung* (BMBF, German Federal Ministry of Research), project "COCOBOTS" (01IS21102A). We thank the anonymous reviewers for their helpful feedback.

## Limitations

As indicated above, the number of instances per experiment is not large. As we did not observe very large deviations in results, we kept the numbers small to reduce (monetary and environmental) cost; acknowledging that larger sets (and tests with different temperature settings) may increase the fine-grainedness of the results. In addition, limited context size may be an issue in models that hallucinate long utterances: if the beginning of the dialogue history gets cropped, the instructions are deleted. We set a maximum number of tokens for the open models. As also discussed above, one limitation that we soon want to overcome is that of a retriction to English language prompts and game instances.

### Limits on reproducibility of closed access models

Some models under evaluation are only accessible via a programming interface which basically adds a black box on top of a black box (GPT-3/3.5/4, Luminous, Claude). The mechanics (and exact models invoked) behind these interfaces might change at any time and consequently the results of successive runs might vary arbitrarily. For the closed models tested here, the best we can do is to provide the timestamp of the testing and the versioning information, to the extent that it is available to us.

### Limits on selection of open access models

The selection of open access models was based on looking at high-ranked models on existing benchmarks (LMSYS, 2023; HuggingFace, 2023) and identifying the candidate ones. Another criterion for the selection was the availability of model weights publicly to ensure the reproducibility of the study.

## Ethics Statement

Using paid proprietary APIs with underlying models about which little is known (training data, model architecture) in academic research is less than ideal. At the moment, the models tested here seem to be the only ones that are even able to follow the structure of the games as instructed. It is our hope that open models will catch up soon, and proper research can be done with them.

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

## A   Detailed Benchmark Results

In this section, we include additional visualisations of the overall results. Figure 10a is a graphical representation of the main results in Table 2. Figure 10c illustrates the percentage of played and aborted games; played games are further split into successful (perfect performance) and lost games. Figure 10b presents the comparison of clemscores for each model.

## B   Common metrics

Besides each game's specific scores, the following metrics are computed for all games:

- **Quality Score**: A custom performance score, normalised to the interval $[0, 100]$, representing the quality of the game play. This is used to compare models across different games, similar

to the preferred score in Srivastava et al. (2022). *Measures: episode performance.*

- **Aborted**: At the episode level, either 0 or 1 whether the game play has been aborted (1) or not (0). A game counts as aborted when a violation of the game rules happens, for example a response is not parsable by the rule that specifies it's format as "*TYPE: <text>*" (or re-prompt for n turns). *Measures: episode performance.*

- **Loss**: At the episode level, either 0 or 1 whether the (non-aborted) game has been successful (0) or not (1). *Measures: episode performance.*

- **Success**: At the episode level, either 0 or 1 whether the (non-aborted) game play has been successful (1) or not (0). *Measures: episode performance.*

- **Request Count**: total number of request given to the model by the GM (usually 1 per turn, but for games with re-prompting this might be >1 per turn). *Measured at: turn and episode level.*

- **Parsed Request Count**: total number of request that could be parsed successfully (the model's response complies to the game rules; accumulates over the episode). *Measured at: turn and episode level.*

- **Violated Request Count**: game master checks the outputted text and decides whether it matches the "game form" (also as a log action), if not then this is a violation of the game rules; total count of failures in a episode; turn-based (can be >= 0). *Measured at: turn and episode level.*

- **Request Success Ratio**: parsing success rate - or prompt has been successful if the output can be parsed properly. It is computed by dividing the parsed request count by the total request count . *Measures: episode performance.*

Together, these scores allow for more fine-grained insights into the performance of the models.

**clemscore**   To facilitate easy comparison of models, we define a score summarising the performance of a model in the benchmark as a whole. The the % of actually played games (*i.e.* not aborted) and the average quality score (over all episodes) are computed for each game, and rounded to two decimals. Then, the macro-average quality score and the

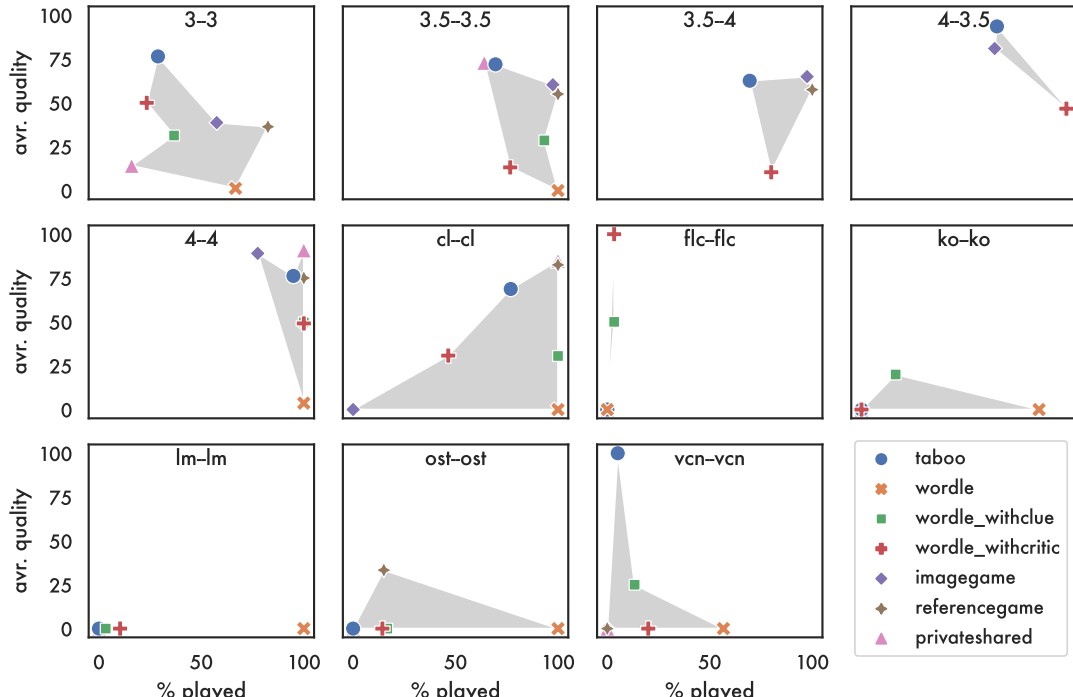

(a) Overview of % played games and micro-average quality score for all models and games. Perfect performance in the benchmark would be represented with all markers overlapping in the top right corner.

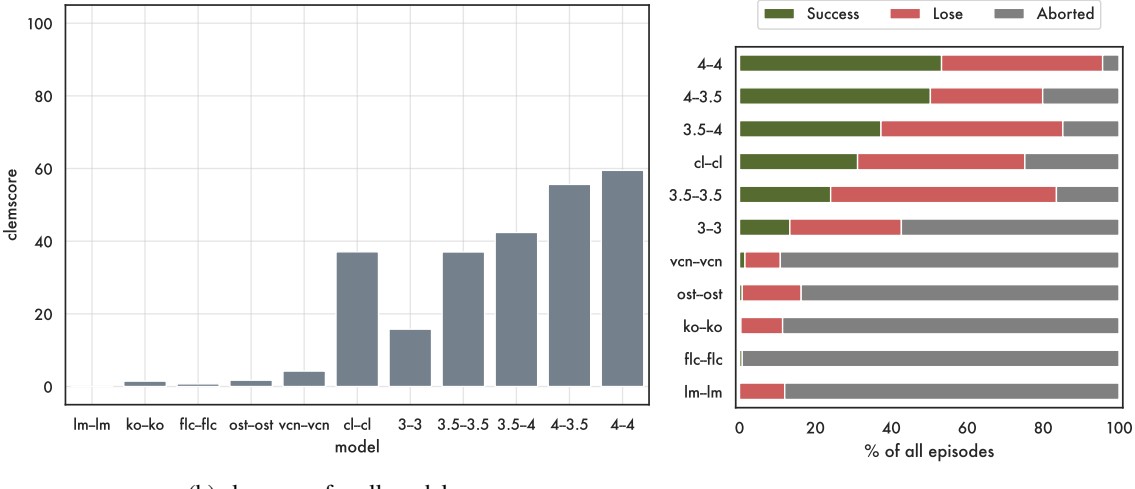

(b) clemscore for all models.

(c) Overall % of successful, lost and aborted games.

Figure 10: Other views on the main results

macro-average % played are computed as the mean over game scores. clemscore is the macro-average quality score multiplied by the macro-average proportion of played games. Given $N$ games, the clemscore of a given model is computed as follows:

$$\left(\frac{1}{N}\sum_{i=1}^{N}q_i\right)\left(\frac{1}{100N}\sum_{i=1}^{N}\%p_i\right)$$

where $\%p_i$ is the percentage of played episodes (*i.e.* episodes that were not aborted) for game $i$, rounded to two decimals, and $q_i$ is the mean qual-

ity score across all game $i$ episodes that were not aborted, rounded to two decimals.

## C    Game: Taboo

### C.1    Game Details

In this game a *Describer* describes a target word for a *Guesser*. The Describer must explain the target word concept without using neither the word itself, nor a number of related words. For example, when the target word is *mark*, the Describer might be told not to use the words *label, tag or stamp*. After each

incorrect guess by the Guesser, the Describer can add to their description. The game ends when the Guesser guesses correctly or a maximum number of turns has been reached.

When the cLLM is playing the Describer, then the game tests its ability to describe concepts and give meaning definitions. In addition, the game tests its helpfulness in the game context: e.g., if a Describer does not alter or extend its initial description after an incorrect guess, we consider this as unhelpful behavior. When playing as a Guesser, then the game tests the cLLM's ability to access its world model. In addition, similarly as above, if a Guesser repeats an earlier guess though given a different description, the model has not aligned well enough to the game goal (has not "understood" the game constraints).

## C.2 Instantiation

The players are each given their own prompts, as shown in Figure 11. We set the maximum number of guesses to 3.

**Target Words.** We use an English word frequency list based on web data (Brants and Franz, 2006)[6] to derive a list of lemmatized target word candidates. From these candidates we remove all that occur less than 5 times per 1 million tokens.

**Frequency-based Experiments.** The remaining candidates are sorted into 3 equally-sized bins based on their frequency in the corpus. The resulting bins can be interpreted as (i) low-frequency words that occur up 9.4 times per 1 million tokens, (ii) the medium-frequency words occur up to 25.1 times per 1 million tokens and (iii) the high-frequency tokens occur up to $1,2951$ times in 1 million tokens. The assumption is that the word level frequency is a proxy for a cLLM's difficulty to describe or understand a word (because it has seen it more or less times during training).

**Game Instances.** From each frequency group we (uniformly) sample 20 words as the target words. We manually ensure that the final word list does not contain inappropriate words such as vulgar language. Then we use the Merriam Webster Thesaurus API to find all synsets for a particular target word. We concatenate the synsets and sample 3 words as the related words. This means that the related words cover a variety of target word meanings. If for some reasons only less than 3 related

[6]https://www.kaggle.com/datasets/rtatman/english-word-frequency

---

```
TEMPLATE C.1.1
You are playing a collaborative word guessing
game in which you have to describe a target
word for another player to guess.

Rules:
(a) You have to reply in the form: CLUE: <some
text>. Guesses from the other player will start
with GUESS.
(b) You cannot use the target word itself,
parts or morphological variants of it in your
description.
(c) In addition, the same rules apply for
related words which are provided below.

End conditions:
(i) If you use the target word or a related
word in your description, then you lose.
(ii) If the other player can guess the target
word in $N$ tries, you both win.

Let us start.

This is the target word that you need to
describe and that the other player needs to
guess:

$TARGET_WORD$

Related words are:

$REL_WORD$

Important: You are under time pressure, give
short descriptions that are to the point!
```

```
TEMPLATE C.1.2
You are playing a collaborative word guessing
game in which you have to guess a target word
that another player describes to you.

You can make one guess at each trial. You win
when you guess the target word. You lose when
you cannot guess it in $N$ tries.

After each trial you will get a new hint from
the other player which starts with CLUE.

Make your guesses by just saying the word using
the following form: GUESS: <a word>

Let us start.
```

Figure 11: The Describer and Guesser prompts for the Taboo game.

words could be chosen via the API, then we manually search for the synonyms words on the Merriam Webster webpage[7] and choose the highest ranked ones.

**Evaluation**    We measure the following metrics at the episode-level:

1. **Success**: Whether or not the Guesser guessed the target word.

2. **Abort**: 1 if any player did not follow the rules, and 0 otherwise.

3. **Speed (Quality Score)**: How early the Guesser guessed the word as measured by $100/t$, where $t$ is the turn number in which the target was found. When the game was unsuccessful, speed is 0. For aborted games, speed is undefined.

**Example transcripts**    We present example transcripts in Figures 12 and 13.

| # | A | Game Master | B |
|---|---|---|---|
| 1 | [A)GM] CLUE: A place where cars and people share the same space. | | |
| 2 | | [GM\|GM] continue | |
| 3 | | [GM\|GM] A place where cars and people share the same space | |
| 4 | [GM)B] CLUE: A place where cars and people share the same space | | |
| 5 | | [GM(B] GUESS: Street | |
| 6 | | [GM\|GM] continue | |
| 7 | | [GM\|GM] street | |

Figure 12: Example of a successful gameplay. The target word was *street*, taboo words were *road, asphalt, drive*.

## C.3 Additional Discussion of Results

The results in Table 2 (see main text) indicate that the open source models Koala, Falcon, OpenAssistant are not able to play the taboo game at all. The same holds for Luminous. The open source Vicuna model plays at least some games and these with 100% success. The performances are better for the GPT-* family of models and Claude.

Here we see Claude is a strong competitor with 76.92% of played games and a quality score (Speed) of 68.75%. These scores exceed the performances of the other pairings (3/3, 3.5/3.5, 3.5/4,

[7]https://www.merriam-webster.com/thesaurus

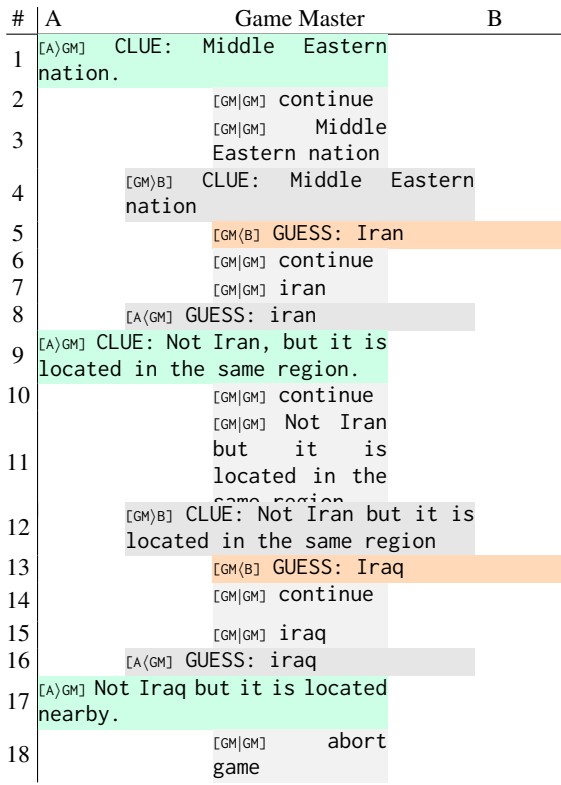

Figure 13: Example of a game that was aborted because the Describer violated the game rules (the description has to begin with CLUE:). The target word was *israel*, taboo words were *country, tel aviv, jew*.

4/3.5) except for GPT-4. The GPT-4 model with self-play is playing almost all games (94.92%) and achieves a high quality score (76.19%). This means that GPT-4 is following the rules of the game in almost all cases.

We hypothesise that this might due to an even longer training with RLHF (and Claude is catching up) so that the model is highly aligned with the prompts given by the user (the game master). An indicator that this hypothesize is justified is given by the jump in games played between GPT-3 (a foundation model) and GPT-3.5 (an RLHF fine-tuned model) from $28.81\% \rightarrow 69.49\%$ while the quality score remains similar between these model. As GPT-3.5 is based on GPT-3 the knowledge representation is shared between these two and both are similarly good in retrieving that knowledge.

Now the pairing of the GPT-3.5 and GPT-4 models show an interesting picture. Both pairings are playing about the same number of games (69.49% vs 66.1%) and as the number is the same (or similar) as the GPT-3.5 self-play results we can argue that the aborted games are due to GPT-3.5. On the other hand we see that the quality score (Speed)

| Model | Experiment | n | Aborted | Played | Speed | Success | Lose |
|---|---|---|---|---|---|---|---|
| **3–3** | **0_high** | 20 | 70.00 | 30.00 | 100.00 | 30.00 | / |
| | **1_medium** | 19 | 78.95 | 21.05 | 75.00 | 15.79 | 5.26 |
| | **2_low** | 20 | 65.00 | 35.00 | 57.14 | 20.00 | 15.00 |
| **3.5–3.5** | **0_high** | 20 | 25.00 | 75.00 | 86.67 | 65.00 | 10.00 |
| | **1_medium** | 19 | 42.11 | 57.89 | 72.73 | 42.11 | 15.79 |
| | **2_low** | 20 | 25.00 | 75.00 | 56.67 | 45.00 | 30.00 |
| **3.5–4** | **0_high** | 20 | 25.00 | 75.00 | 74.44 | 65.00 | 10.00 |
| | **1_medium** | 19 | 42.11 | 57.89 | 68.18 | 42.11 | 15.79 |
| | **2_low** | 20 | 25.00 | 75.00 | 46.67 | 40.00 | 35.00 |
| **4–3.5** | **0_high** | 20 | 30.00 | 70.00 | 96.43 | 70.00 | / |
| | **1_medium** | 19 | 26.32 | 73.68 | 100.00 | 73.68 | / |
| | **2_low** | 20 | 45.00 | 55.00 | 81.82 | 45.00 | 10.00 |
| **4–4** | **0_high** | 20 | 10.00 | 90.00 | 83.33 | 85.00 | 5.00 |
| | **1_medium** | 19 | / | 100.00 | 71.93 | 84.21 | 15.79 |
| | **2_low** | 20 | 5.00 | 95.00 | 73.68 | 75.00 | 20.00 |
| **cl–cl** | **0_high** | 14 | 21.43 | 78.57 | 77.27 | 78.57 | / |
| | **1_medium** | 18 | 22.22 | 77.78 | 75.00 | 66.67 | 11.11 |
| | **2_low** | 20 | 25.00 | 75.00 | 56.67 | 50.00 | 25.00 |
| **flc–flc** | **0_high** | 20 | 100.00 | / | / | / | / |
| | **1_medium** | 19 | 100.00 | / | / | / | / |
| | **2_low** | 20 | 100.00 | / | / | / | / |
| **ko–ko** | **0_high** | 20 | 100.00 | / | / | / | / |
| | **1_medium** | 19 | 100.00 | / | / | / | / |
| | **2_low** | 20 | 100.00 | / | / | / | / |
| **lm–lm** | **0_high** | 20 | 100.00 | / | / | / | / |
| | **1_medium** | 19 | 100.00 | / | / | / | / |
| | **2_low** | 20 | 100.00 | / | / | / | / |
| **ost–ost** | **0_high** | 20 | 100.00 | / | / | / | / |
| | **1_medium** | 19 | 100.00 | / | / | / | / |
| | **2_low** | 20 | 100.00 | / | / | / | / |
| **vcn–vcn** | **0_high** | 20 | 90.00 | 10.00 | 100.00 | 10.00 | / |
| | **1_medium** | 19 | 94.74 | 5.26 | 100.00 | 5.26 | / |
| | **2_low** | 20 | 100.00 | / | / | / | / |

Table 3: Overview of the models performances for all experiments in Taboo. The cells are color coded with traffic light colors so that green means high performance and red means low performance.

jumps over 31.09 scores (from 62.6% to 93.59%). This shows that the GPT-4 model is a better Describer than GPT-3.5 and it is a better prompter for "knowledge retrieval" than GPT-3.5 (with a quality score of 71.95%).

Still, especially the number of games played (without a rule violation) is less than what we would expect from a human player. We will test human abilities to play this game in a future iteration using slurk.

**Effect of word frequency on model performance.**
Figure 14a shows that the word frequency of the target words has no clear effect on the number of games played for the models. But we can see in Figure 14b that the frequency indeed impacts the quality score (Speed) of the models: with a lower frequency the models have a harder time to find the correct word to be guessed. This is reasonable – but also a bit counter-intuitive as these models are expected to have enough capacity to store everything – because when a word is seen in more contexts during training, then the Describer has a better chance (a) to either prompt for the context that is most often seen and thus has been

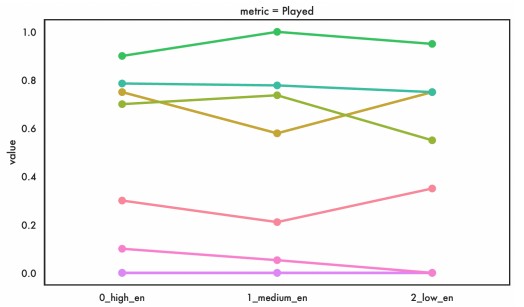

(a) For each model the mean number of games played (by experiment).

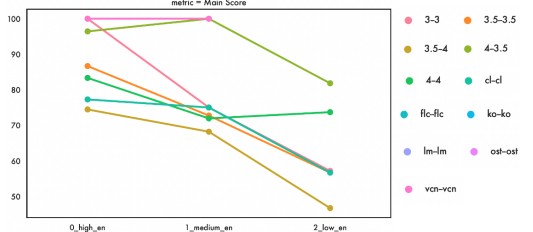

(b) For each model the mean main scores (Speed) (by experiment).

Figure 14: Two important performance indicators for the models separated by experiment.

more manifested in the model's weights or (b) can prompt in various ways for the target word (probing the Guesser for the knowledge). Detailed results are given in Table 3.

## D    Game: Wordle

The popular word guessing game "Wordle" gained global attention, in which players are challenged to guess a five-letter word in six attempts. After each guess, the player receives feedback indicating which letters are in the correct position, which letters are correct but in the wrong position, and which letters are incorrect, to help them strategise their next guess. The objective of the game is to guess the target word using the fewest possible guesses, and the game ends when the player guesses correctly or exhausts all six attempts.

### D.1    Game Details

**Wordle (Traditional Variant)**    This game evaluates three key aspects of cLLM's capabilities. Firstly, it assesses how well the cLLM comprehends the game rules, which involves generating valid English words consisting of exactly five letters. Secondly, it measures how effectively cLLM uses guess feedback to generate its next guesses. Thirdly, it measures how quickly cLLM can guess the target word if it succeeds.

In traditional gameplay, cLLM plays the role of "Player A", and a deterministic wordle bot plays the role of "Player B". Game begins with the game master prompting Player A to guess the target word. The game master parses Player A's response and forwards it to Player B, which evaluates the closeness of the guess word to the target word and returns the feedback. The game master sends the feedback to Player A for the next guess and the cycle continues until the target word is guessed correctly or all six attempts are exhausted. The prompt template of this variant is available in Figure 15a.

**Wordle (+ Semantics-Based Clue)**    This is a Wordle variant where the guesser (Player A) gets a clue before starting to guess. For example, for the target word PRIDE, the clue could be "pack of lions". The rest of the game rules follow the same as the traditional game variant. cLLM plays the role of the "player A", and a deterministic wordle bot plays the role of "player B".

The primary aim of testing this variant is to evaluate the efficacy of Player A in effectively utilising the supplementary information provided by a clue to improve its guess of the target word. The clue serves as an aid to narrow down the possible word options. The success of the game depends on Player A's ability to integrate the clue with the guess_feedback. Player A's explanation offers insights into how the cLLM links the clue phrase and the guess_feedback. The prompt template is available in Figure 15b.

**Wordle (+ Clue, + Critic)**    This game variant also begins with the guesser (Player A) who attempts to guess the target word based on a given clue. In contrast to other game variants, where the guessed word is immediately evaluated for its proximity to the target word, in this variant, the guessed word and the clue are forwarded to another player known as the *critic*, to get an opinion on the correctness of the guess. The critic responds with either agreement or disagreement, providing their rationale based on the information given. The critic's response is then relayed to the guesser, who can decide to stick with their initial guess or change it based on the feedback received. Figure 16a shows the prompt structure for the Player A, Figure 16b shows the prompt structure for the critic role and Figure 18 depicts the prompts fed to the guesser to share the critic's opinion.

This game variant helps to investigate the in-

TEMPLATE D.1.1
You are a language wizard who likes to guess words by using the given rules.

Welcome to Wordle! You have six attempts to guess the target word, a valid English word of five lowercase letters (a-z). Please use the tags "guess:" and "explanation:" to provide a concise explanation for each guess.

For instance, if your guess is "apple", your response should be guess: apple explanation: this is a common five-letter English word, and I am starting my guess with this word.

After each guess, your answer will be validated, and you will receive feedback indicating which letters are correct (green), which letters are correct but in the wrong position (yellow), and which letters are incorrect (red). This feedback can be useful in determining which letters to include or exclude in your next guess.

For example, the feedback for "apple" might be:
guess_feedback:
a ⟨yellow⟩ p ⟨yellow⟩ p ⟨green⟩ l ⟨yellow⟩ e ⟨red⟩

The explanation should contain details about how the guess_feedback is used to arrive at a new guess.

Let's begin with your first guess.

---

TEMPLATE D.1.2
guess: hello
explanation: This is a common five-letter English word, and I am starting my guess with this word.

---

TEMPLATE D.1.3
You are a language wizard who likes to guess words by using the given rules.

Welcome to Wordle! You have six attempts to guess the target word, a valid English word of five lowercase letters (a-z). Please use the tags "guess:" and "explanation:" to provide a concise explanation for each guess.

To help you make an informed guess, you will receive a clue for the word, such as clue: snowy white.

Here is an example guess based on the clue: guess: apple
explanation: In the fairy tail Snow White, the girl is killed because she eats a poisoned apple. And the word apple has 5 letters.

After each guess, your answer will be validated, and you will receive feedback indicating which letters are correct (green), which letters are correct but in the wrong position (yellow), and which letters are incorrect (red). This feedback can be useful in determining which letters to include or exclude in your next guess.

For example, the feedback for "apple" might be:
guess_feedback:
a ⟨yellow⟩ p ⟨yellow⟩ p ⟨green⟩ l ⟨yellow⟩ e ⟨red⟩

The explanation should contain details about how both the guess_feedback and clue work together in tandem to formulate a new guess.

Let's begin with your first guess.
clue:display

---

TEMPLATE D.1.4
guess: screen
explanation: A display can refer to a computer screen, and "screen" is a valid English word with five letters.

(a) Player A (Guesser) in the Wordle Game Basic Variant

(b) Player A (Guesser) in the Wordle Game Variant (+Clue)

Figure 15: Wordle prompt templates for basic and with clue variants

TEMPLATE D.1.5
You are a language wizard who likes to guess
words by using the given rules.

Welcome to Wordle!  You have six attempts
to guess the target word, a valid English word
of five lowercase letters (a-z). Please use the
tags "guess:" and "explanation:" to provide a
concise explanation for each guess.

To help you make an informed guess, you
will receive a clue for the word, such as
clue: "snowy white"

Here is an example guess based on the
clue:
guess: apple
explanation: In the fairy tail Snow White, the
girl is killed because she eats a poisoned
apple. And the word apple has 5 letters.

I will then indicate whether I agree or
disagree with your guess and provide rationale,
but agreeing with a guess does not confirm its
correctness.  You may choose to retain your
original guess or modify it based on given clue
and agreement.

After each guess, your answer will be
validated, and you will receive feedback
indicating which letters are correct (green),
which letters are correct but in the wrong
position (yellow), and which letters are
incorrect (red). This feedback can be useful
in determining which letters to include or
exclude in your next guess.

For example, the feedback for "apple"
might be:
guess_feedback:
a ⟨yellow⟩ p ⟨yellow⟩ p ⟨green⟩ l ⟨yellow⟩ e
⟨red⟩

Here are some rules to keep in mind: 1.
Please do not generate any guess feedback or
clue on your own 2. Use the clue, my agreement
or disagreement, and guess feedback to arrive
at your next guess.  This should be reflected
in your explanation as well.

Let's begin with your first guess.
clue:display

TEMPLATE D.1.7
I need your assistance with a word game in
which we need to find a 5-letter word using
a clue, a guess and an explanation for the guess.

For example, if the clue is "snow white"
and the guess is "apple", you will get the
information like this:
clue:snow white
guess:apple
explanation:the fairy tale Snow White involves
the girl eating a poisoned apple, which has 5
letters.

Your task is to either agree or disagree
with my guess based on the given clue.

If you disagree, I will make another guess, and
if you agree, I will proceed in the game using
the same guess.

Initially, guess feedback is not available,
but as the game proceeds, it becomes available.
The guess feedback is not for current guess,
but rather an indication of what letters can
be present in the current guess.  A letter
highlighted in green indicates that it is a
correct letter in the correct position, while
yellow indicates that it is a correct letter
in the wrong position, and red indicates a
wrong letter.  At this point, you should use
all the provided information, including the
clue, guess, explanation, and guess feedback,
to decide whether the given guess matches the
clue, and also stick to guess feedback or not.

Please respond in lowercase letters and
stick to this format:
agreement:yes or no
explanation:your reason for doing so

Please note that you must not guess any
word based on the riddle clue and stick to the
given format while responding.

Let's begin.

clue:display
guess:shelf
explanation:A shelf is a common item used for
displaying objects. It has 5 letters.

TEMPLATE D.1.6
guess: shelf
explanation: A shelf is a common item used for
displaying objects. It has 5 letters.

TEMPLATE D.1.8
agreement: no
explanation: None of the letters in "shelf"
match with the letters that could be present in
the word based on the given guess feedback.

(a) Prompt template for Player A (Guesser) in the Wordle Game Variant (+Clue, +Critic)

(b) Prompt template for Player B (Critic) in the Wordle Game Variant (+Clue, +Critic)

Figure 16: Wordle prompt templates for players with clue and critic variants

fluence of the critic's role in the guesser's performance and can lead to interesting possibilities in human-machine interaction, where the human can be aided by the cLLM as the critic. We tested the game using the same cLLM for both roles, as well as different cLLMs for each role, employing distinct prompts for each.

**Instantiation** In our experiments, we use a list of 2,309 possible target words and a list of 12,953 valid guess words.[8] For textual clues, we use New York Times crossword clues.[9] We sort the target words by word frequency.[10] Out of the initial 2,309 target words, frequency details are not available for one word, and clues are not available for 39 words. These words are subsequently excluded from the experiments. The remaining 2,269 target words are sorted based on their word frequency (descending frequency) and then divided into three equal groups. The first group which contains high-frequency words, has a total of 756 words. The second group, consisting of words with medium frequency, also contains 756 words. Finally, the third group, which contains low-frequency words, has a total of 757 words. To evaluate our methodology, we chose (random seed: 42) 10 words from each frequency group, resulting in a total of 30 target words for evaluation purposes, for each game variant. As metrics, we keep track of the success rate (how often the guesser guessed the target word, within the limit of 6 guesses), the average speed (if successful, then at which turn), and for each turn closeness (based on the letter-feedback). We also keep track of whether the guesser repeats a guess (a strategic failure), and, in the critic variant, whether the guesser changes the guess after feedback.

**Error Handling** The experiments revolve closely around the cLLM models, which are expected to respond in a specific format and adhere to certain rules. However, there are multiple scenarios where the responses from these models may result in errors.

1. In the Wordle game, a subset of valid five-letter English words is used. In certain scenarios, the guesser (Player A - cLLM) may guess

a valid 5-letter word that is not among the allowed guesses. In such cases, cLLM will be asked to guess another word. This reprompting process continues until cLLM makes an allowed guess.

2. The Wordle game has a strict rule that allows guessing only 5-letter words. Sometimes, the models respond with words that do not adhere to this restriction, causing the reprompting. We allow two reprompting attempts, after which the game is considered aborted.

3. Sometimes, the response of the cLLM doesn't follow the expected format as stated in the prompt. In such cases, we reprompt the cLLM to generate the response in the expected format. When faced with these circumstances, we usually give two reprompts before declaring the game as aborted.

**Evaluation** For each episode, we record the number of guesses made by the guesser. If the guesser correctly guessed the word in six or fewer attempts, the game is counted as a success. If the guesser exhausted all six attempts, the game is counted as a failure. If the guesser's response does not conform to the game rules, the game is counted as aborted. Of the successful games, the average number of guesses taken to guess the word is computed. For all the games, we also measured how close the guess gets to the target word with each turn. The following are the metrics measured for each episode.

1. **Success**: This is a binary value and measures whether the guesser guessed the target word or not.
2. **Aborted**: This is a binary value and measures whether the game aborted due to non-compliance with the game rules (words not containing 5 letters, words containing symbols other than alphabets).
3. **Speed**: How early the word was guessed as measured by $100/t$, where $t$ is the turn number in which the target was found.
4. **Closeness**: This contains the score ranging from 0-to-25 and determines how effectively the guesser utilizes the guess feedback. If a letter is at the correct position 5-points are awarded, and 3-points for letter at other position and 0-points for incorrect letters, leading to 25 points for a correct guess. Ideally this score should be increase across the turns.

---

[8] https://github.com/3b1b/videos/blob/master/_2022/wordle/data/allowed_words.txt https://github.com/3b1b/videos/blob/master/_2022/wordle/data/possible_words.txt

[9] https://www.kaggle.com/datasets/darinhawley/new-york-times-crossword-clues-answers-19932021

[10] https://www.kaggle.com/datasets/rtatman/english-word-frequency

| wordle | Played | Aborted | Success | Lose | Speed |
|---|---|---|---|---|---|
| lm–lm | **1.00** | **0.00** | 0.00 | 1.00 | 0.00 |
| ko–ko | 0.87 | 0.13 | 0.00 | 0.87 | 0.00 |
| flc–flc | 0.00 | 1.00 | 0.00 | 0.00 | UNDEF |
| ost–ost | **1.00** | **0.00** | 0.00 | 1.00 | 0.00 |
| vcn–vcn | 0.57 | 0.43 | 0.00 | 0.57 | 0.00 |
| cl–cl | **1.00** | **0.00** | 0.00 | 1.00 | 0.00 |
| 3–3 | 0.67 | 0.33 | 0.03 | 0.63 | 1.25 |
| 3.5–3.5 | **1.00** | **0.00** | 0.00 | 1.00 | 0.00 |
| 4–4 | **1.00** | **0.00** | **0.23** | 0.77 | 3.67 |
| **wordle + clue** | **Played** | **Aborted** | **Success** | **Lose** | **Speed** |
| lm–lm | 0.03 | 0.97 | 0.00 | 0.03 | 0.00 |
| ko–ko | 0.17 | 0.83 | 0.03 | 0.13 | 20.00 |
| flc–flc | 0.03 | 0.97 | 0.03 | 0.00 | 50.00 |
| ost–ost | 0.17 | 0.83 | 0.00 | 0.17 | 0.00 |
| vcn–vcn | 0.13 | 0.87 | 0.03 | 0.10 | 25.00 |
| cl–cl | **1.00** | **0.00** | 0.47 | 0.53 | 30.56 |
| 3–3 | 0.37 | 0.63 | 0.20 | 0.17 | 31.36 |
| 3.5–3.5 | 0.93 | 0.07 | 0.27 | 0.67 | 28.57 |
| 4–4 | **1.00** | **0.00** | **0.73** | 0.27 | 49.67 |
| **wordle + clue + critic** | **Played** | **Aborted** | **Success** | **Lose** | **Speed** |
| lm–lm | 0.10 | 0.90 | 0.00 | 0.10 | 0.00 |
| ko–ko | 0.00 | 1.00 | 0.00 | 0.00 | UNDEF |
| flc–flc | 0.03 | 0.97 | 0.03 | 0.00 | 100.00 |
| ost–ost | 0.14 | 0.86 | 0.00 | 0.14 | 0.00 |
| vcn–vcn | 0.20 | 0.80 | 0.00 | 0.20 | 0.00 |
| cl–cl | 0.46 | 0.54 | 0.14 | 0.32 | 30.77 |
| 3–3 | 0.23 | 0.77 | 0.13 | 0.10 | 50.00 |
| 3.5–3.5 | 0.77 | 0.23 | 0.2 | 0.57 | 13.19 |
| 4–4 | **1.00** | **0.00** | **0.8** | 0.2 | 49.11 |

Table 4: Detailed results for the wordle games (traditional, clue, critic variants).

## D.2   Additional Discussion of Results

The detailed results for all three variants of the wordle game is given in Table 4. In terms of overall performance, the best model is GPT-4, followed by GPT-3.5 and Claude, with GPT-3 following after them.

GPT-4 is the only model that can always follow the game rules (*Played*) in all 3 variants. While its performance for the traditional wordle game is relatively low with a success rate of 0.23, this score is greatly increased by adding a clue (0.73) or a critic (0.8). Likewise, the speed metric is increased from 3.67 to 49.67 and 49.11, respectively, meaning that on average the model can find the target word on the second guess in these settings.

For the other models, the regular wordle game seems to be too difficult to play, i.e. following only letter-based feedback is too difficult as the high *Lose* numbers show. Except for Falcon, they can however follow the game rules (cf. *Played*) in at least half of the episodes.

For several models, the *Played* score decreases in the extended games variants (clue; clue+critic). The game rules seem to be too difficult for Luminous, Koala, Vicuna, and GPT-3 that drop to scores

0.03, 0.17, 0.13, 0.37 in the clue variant and 0.10, 0.00, 0.20, 0.23 in the clue+critic variant. For GPT-3.5, the drop is smaller (from 1.00 to 0.93 and 0.77, respectively).

Because of the high number of aborted episodes, we present results for *Closeness* only for the GPT-4 model in Figure 17. The figure shows the closeness score for all episode that GPT-4 has played, grouped by game variant and word frequency. It appears that the word frequency may have an effect in the case of the extended game variants: The higher the word frequency, the more stable the progression towards the target seems to be. This figure also reflects the speed: We can see that adding a clue and critic results in the model being able to guess the target word correctly on the first attempt (indicated by 'circle' markers in the plots) in multiple episodes.

**Comparison with Human Performance**   While we were unable to find actual playing statistics, there are plenty of blog posts advising players on playing strategies. The New York Times (who is hosting the official game version) is suggesting the

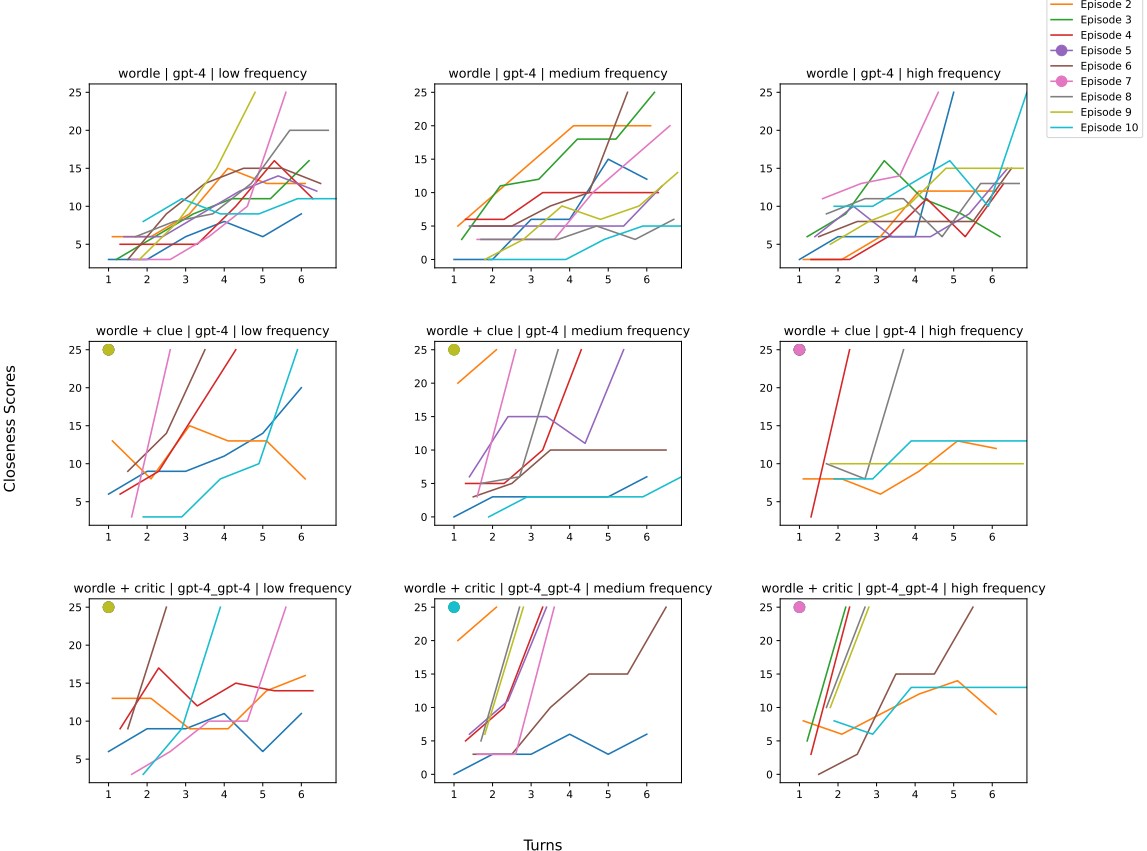

Figure 17: Closeness Scores Progression for all episodes of GPT-4 play

following strategies[11] that we can compare to the models' actions:

- *Start with the same word in every game or start with an entirely new word each time.* Each of the models follows the first pattern of always starting with the same word. In the regular wordle setting, all models except Claude and GPT-3.5 always start with `apple` as their first guess. GPT-3.5 always starts an episode with `hello` as first guess. Claude mostly starts with `apple` and sometimes starts with `crane` (4/30 times).

- *Start with as many vowels as possible.* This translates into "words that have 3 or more vowels". This is not a strategy that cLLM's follow as the previous point showed. Neither `hello` nor `apple` have particularly many vowels.

- *Cover many different letters in the first two guesses.* This entails disregarding the letter-based feedback of the first guess in favor of trying many letters. We cannot see that this happens

in the second guess. On the contrary, letters are often repeated and for most models, the second guess is always the same guess, e.g., GPT-3.5 almost always guesses first `hello` and then `world`. The only model that guesses a different word over all second guesses is GPT-4. The claude model is fairly varied with 5 different words over the 30 episodes. All models however repeat letters from the first guess, i.e. they do not follow this strategy.

- *Use a database of words.* This is meant to help with picking words when there are several options so that the player can choose one that eliminates many others. While there is no explicit database contained in cLLMs, we still expect the models to be more aware of possible option than humans would be with their limited retrieval capacity.

- *Remember there can be duplicate letter.* This does not seem to be an issue for the models in the sense that they are not avoiding it. We often see *hello* as a guess. On the contrary, models

---

[11] https://www.nytimes.com/2022/02/10/crosswords/best-wordle-tips.html

```
TEMPLATE D.2.9
clue:display
guess_agreement:no
agreement_explanation:None of the letters in
"shelf" match with the letters that could be
present in the word based on the given guess
feedback.
```

```
TEMPLATE D.2.10
guess:stand
explanation: A stand is an object used for
displaying items. It has 5 letters.
```

Figure 18: Wordle prompt template for Player A (Guesser) to share critic's opinion in the Wordle Game Variant (+Clue, +Critic)

do not seem to have the capacity to guess by exclusion, i.e. use distinct letters as a strategy to find correct ones.

- Other tips include *use pen and paper* and *use your guessing stats to inform your strategy*, neither of which are applicable here since the cLLMs do not keep any memory of previous episodes.

**Probing for Explanations** As mentioned in the previous paragraph, most models are fairly consistent in their choice of guess and use the same words in every episode for the first two guesses, e.g., GPT-3.5 almost always guesses first `hello` and then `world`. The only model that guesses a different word over all second guesses is GPT-4.

During the game play, we also ask the models to explain their guesses. We examine these explanations here to get insight into whether GPT-4 can correctly translate the feedback into a next guess.

We annotate the subset of the regular wordle episodes played by the GPT-4 and GPT-3.5 models as follows, using only the high-frequency word episodes (cf. Appendix D.1). Examples for the labels can be found in Table 5:

- EXPL-INCORRECT – In this turn, the explanation is incorrect with regard to the previous guess or guesses.

- CONCL-INCORRECT – In this turn, the explanation is correct with respect to the previous guesses, but the conclusion is incorrect, inconsistent, or incomplete. The guess itself counts as part of the conclusion as well.

- GOOD – This turn constitutes a good guess, i.e. the feedback was correctly explained, the conclusion drawn from it is correct (even with respect to other previous guesses), and the guess adheres to explanation and conclusion.

Each guess comes with an explanation, but we do not count the players' first guess towards the total, because there is no previous letter-based feedback available. The models always justify their first guess with a variant of "This is a common five-letter word."

We take a strict approach, expecting a player to adhere to *all* parts of the feedback and be able to remember *all* of their previous guesses. We count a feedback explanation as correct if all green and yellow letters are mentioned. Red letters may stay implicit.

Both GPT-3.5 and GPT-4 have perfect *Played* score for the traditional wordle game (cf. Table 4). GPT-3.5 was not able to win any episode, GPT-4 won only a small number (23%). The analysis of 50 guess explanations for the GPT-3.5 model and 47 for the GPT-4 model reveals that GPT-3.5 is hardly able to handle the letter-based feedback at all: 52% of the explanations are labeled EXPL-INCORRECT (for examples see Table 5). GPT-4 was much better at explaining the letter-based feedback as only 4% of the explanations were incorrect (EXPL-INCORRECT). When it comes to the conclusion (i.e. when the explanation was correct), both models struggle to incorporate all aspects of the game rules: GPT-3.5 fails to draw a correct, consistent, and complete conclusion in 46% of the 50 turns and can in fact only generate a single GOOD turn. GPT-4 manages to generate a GOOD guess in about a quarter of the turns (26%). The remaining 70% of the 47 turns are labeled as CONCL-INCORRECT, i.e. there is plenty of room to improve.

## E  Game: Drawing Instruction Giving and Following

### E.1  Game Details

In this game, the task is to draw a simple grid where Player A (Instruction Giver) needs to instruct Player B (Instruction Follower) what to draw, starting from an empty grid. The Game Master instructs the Player A to generate a drawing instruction that applies to the given *target grid*. The expression could include phrases that refer to the

| # | A | Game Master | B |
|---|---|---|---|
| 1 | [A⟩GM] guess: hello
explanation: This is a common five-letter English word, and I am starting my guess with this word. | | |
| 2 | | [A⟨GM] guess_feedback: h<red> e<yellow> l<yellow> l<red> o<yellow> | |
| 3 | [A⟩GM] guess: world
explanation: The feedback indicates that the letters "e" and "o" are correct but in the wrong position. The word "world" has these letters in the correct positions, so I am guessing this word. | | |
| 4 | | [A⟨GM] guess_feedback: w<red> o<green> r<red> l<yellow> d<yellow> | |
| 5 | [A⟩GM] guess:swirl
explanation: The feedback indicates that the letters "o" is correct and in the correct position. The letter "l" is correct but in the wrong position. The word "swirl" has these letters in the correct positions, so I am guessing this word. | | |
| 6 | | [A⟨GM] <continues for remaining attempts> | |
| 7 | | [GM\|GM] attempts: 6
target_word = model
guess: clerk
guess_feedback:  c<red> l<yellow> e<yellow> r<red> k<red> | |
| 8 | | [GM\|GM] game_result = LOSS | |

(a) Traditional variant without clue and critic

| # | A | Game Master | B |
|---|---|---|---|
| 1 | [A⟩GM] guess: rigid
explanation: The clue "unbending" suggests a word that means inflexible or stiff. "Rigid" fits this description and has five letters. | | |
| 2 | | [A⟨GM] guess_feedback: r<red> i<yellow> g<red> i<red> d<red> | |
| 3 | [A⟩GM] guess:crisp
explanation: The feedback shows that none of the letters in "rigid" are correct. The clue "unbending" could also describe something that is firm or fresh. "Crisp" fits this description and has five letters. | | |
| 4 | | [A⟨GM] guess_feedback: c<red> r<red> i<green> s<yellow> p<red> | |
| 5 | [A⟩GM] guess:spicy
explanation: The feedback shows that the second letter is correct and in the right position. The clue "unbending" doesn't seem to be a direct clue for "spicy," but it could be interpreted as something that is strong or intense. "Spicy" fits this description and has five letters. | | |
| 6 | | [A⟨GM] <game flow continues> | |
| 7 | | [GM\|GM] attempts: 6
target_word = stiff
target_word_clue = unbending
guess: split
guess_feedback:  s<green> p<red> l<red> i<yellow> t<yellow> | |
| 8 | | [GM\|GM] game_result = LOSS | |

(b) Variant with only clue

Figure 19: Excerpts of *wordle* game plays for the traditional variant (a) and the one with only clue (b) (GPT-3.5/GPT-3.5)

| # | A | Game Master | B |
|---|---|---|---|
| 1 | [A⟩GM] guess: crook
explanation: The word "uneven" makes me think of something that is not straight or has a bend, like a crook. | | |
| 2 | | [GM\|GM] Share the guess information to critic | |
| 3 | | | [GM⟨B] agreement: no
explanation: The guess "crook" has 5 letters, but the word "uneven" does not necessarily imply a bend or a curve. It could also mean something that is not equal or balanced. |
| 4 | | [GM\|GM] Critic disagrees with the Guesser – Sharing the critic's explanation with the guesser | |
| 5 | [A⟨GM] relays critic's opinion to the guesser | | |
| 6 | [A⟩GM] guess: lumpy
explanation: When I think of "uneven," I think of something that is not smooth or flat, like a lumpy surface. The word "lumpy" has 5 letters. | | |
| 7 | | [GM\|GM] Change in player1's guess
guess_before_critic_opinion: crook
critic_agreement: no
guess_after_critic_opinion: lumpy
Proceeding with guess validation | |
| 8 | [A⟨GM] guess_feedback: l<red> u<red> m<red> p<red> y<green> | | |
| 9 | [A⟩GM] guess:gravy
explanation: The word "lumpy" makes me think of a food that has lumps in it, like gravy. The word "gravy" has 5 letters. | | |
| 10 | | [GM⟩B] clue:uneven
guess:gravy
explanation:The word "lumpy" makes me think of a food that has lumps in it, like gravy. The word "gravy" has 5 letters.
guess_feedback: l<red> u<red> m<red> p<red> y<green> | |
| 11 | | | [GM⟨B] agreement: no
explanation: The guess "gravy" does not match the clue "uneven". Additionally, the guess feedback does not contain any letters that match the letters in "uneven". |
| 12 | | [GM\|GM] Critic disagrees with the Guesser – Sharing the critic's explanation with the guesser | |
| 13 | | [GM\|GM] 'guess:': 'crinkl', 'explanation:': 'When I think of "uneven," I think of something that is not smooth or flat, like a crinkled surface. The word "crinkl" is not a valid English word, so I will try another guess.' | |
| 14 | | [GM\|GM] Guesser Error: INVALID_WORD_LENGTH while parsing Player 1's (model: gpt-3.5-turbo) response, retrying | |
| 15 | | [GM\|GM] Aborted due to invalid format in response | |
| 16 | | [GM\|GM] game_result = ABORTED | |

Figure 20: Excerpt of *wordle* game play for the variant with clue and critic (GPT-3.5/GPT-3.5)

| Label | Context | Examples | Comment |
|---|---|---|---|
| EXPL-INCORRECT | H E L L O | guess:world. explanation: The feedback indicates that the letters *e* and *o* are correct but in the wrong position. The word *world* has these letters in the correct positions, so I am guessing this word. | The explanation fails to mention the misplaced l |
| EXPL-INCORRECT | H E L L O | guess:mango. explanation: Since the feedback shows that the letter *o* is in the correct position, I will try to use it in my next guess. Mango is a common five-letter English word that has *o* in it. | The explanation incorrectly says that o is in the correct position. |
| EXPL-INCORRECT | H E L L O / W O R L D | guess:happy. explanation: Since the feedback shows that none of the letters in *world* are correct, I will try a new word. *Happy* is a common five-letter English word that shares one letter with *world* in the correct position. | The explanation incorrectly says that none of the letters in the previous guess are in the target word when actually l was tagged as yellow, i.e. as being in the target word but in another position. |
| CONCL-INCORRECT | H E L L O | guess:world. explanation: Since the feedback shows that the second and fifth letters are correct but in the wrong position, I will try a word that has *o* and *l* in it, but in different positions. *World* fits this criteria. | The letter-based explanation is correct but the conclusion then suggests to use l which was tagged as not being present in the target word. |
| CONCL-INCORRECT | H E L L O / W O R L D / H A P P Y / S M I L E | guess:grape. explanation: Since all the letters in my previous guess were incorrect, I will try a completely different word. *Grape* is a common five-letter English word that has not been guessed yet. | The letter-based feedback is correct, as is the statement that grape has not been guessed before. However, the new guess contains the letters a and e that have already been tagged as not present in the target word. |
| GOOD | A P P L E | guess:agree. explanation: Based on the feedback, 'a' and 'e' are in the correct positions, and 'p' and 'l' are not in the word. I am guessing *agree* as it is a common five-letter word with *a* and *e* in the correct positions. | The letter-based feedback is correct, the conclusion is correct, and the actual guess keeps the green a and e in the same positions and uses new letters for the other positions. |

Table 5: Examples for labels in the wordle explanations. Context is the player's previous guess and their feedback. The feedback is visualized with colors. The text is formatted for readability.

group of cells in a certain pattern or a single cell and includes additional information about the letter to be filled with. The Game Master passes the generated instruction to the Player B and instructs it to draw the grid that matches the given expression. In the first turn, the Player B starts initialises a grid with empty cells. An empty cell is indicated by the character "□", and a filled cell is an occurrence of any uppercase letter in the alphabet. The Player B applies the given expression to the current state of the grid and returns the result after each turn. The Player A continues to generate expressions until the filled cells in the target grid are described and the Player B keeps updating the current grid incrementally throughout the played turns in the game. The game finishes when Player A generates "DONE". As a fallback, the game also stops when the number of turns reaches the total number of cells in the target grid. The prompt templates for both players are given in Figure 23.

**Instantiation** We experiment with two different settings for *datasets* in this game called *compact* and *random* grids. Each dataset includes 20 different grids resulting in a total of 40 grids, which are 5x5. A **compact grid** stands for a grid with filled cells that follow a certain pattern. Ideally, such grids can be filled by describing the pattern in a single turn or less number of turns than by describing each filled cell one at a time. Each target grid includes at least five filled cells with the same letter (randomly selected for each instance). We manually defined 20 grids that have certain patterns, e.g. filled as M, cross, two rows are filled, three columns are filled, etc. A **random grid** is a randomly initialised grid where the cells generally do not follow a certain pattern when filled. Each target grid includes at least five and at most ten filled cells with the same letter (randomly selected for each instance). The location of each cell is randomly selected.

The main idea for having two different datasets is to test whether the evaluated language models can generate instructions that are compact (Player A side) and whether the generated instruction can be executed to obtain the drawing of the target grid (Player B side). Also, testing with random grids may reveal whether the game can be played with multiple turns by describing each filled cell one turn at a time.

**Evaluation** The evaluation of each episode is carried out by calculating three different measurement types.

1. **Target $\longleftrightarrow$ Drawn grid**: The comparison is done by comparing each filled cell in the target grid with the one at the same position in the drawn grid and calculate *Precision*, *Recall* and *F1-score*. At the turn level, we calculate these scores given the drawn grid up to that point. At the episode level, the drawn grid at the last turn is used. So the *incremental behaviour is to see an increase* in the scores after each interaction.

2. **Changed cell count**: We keep track of the number of cells that change after applying the given instruction on the Player B side. It reveals how certain generated expressions lead to the change of multiple cells, which can be an indication of *compact* instructions. At the turn level, it is simply the number of changed cells in the current state of the grid (after applying the instruction in the turn) with a comparison to the previous state of the grid. At the episode level, the number of changed cells at each turn is averaged.

3. **Generated instruction length**: it measures the number of characters in the generated instruction by the Player A at each turn. At the episode level, it is the average of number of characters in the generated instructions at each turn.

4. **Generated instruction token size**: it measures the average number of tokens in the generated instruction by the Player A at each turn. At the episode level, it is the average of number of characters in the generated instructions at each turn.

**Example transcripts** We present example transcripts for both compacts and random grids in Figure 21a, Figure 21b, Figure 22a, Figure 22b.

## E.2 Additional Discussion of Results

### E.2.1 Overall Comparison

A closer look at the results (see Table 6) reveals that *4–4* is the best model in terms of reaching the highest *success rate* and *F1 score* for both experiments with *random* and *compact* grids. Each experiment is composed of 20 instances and the presented metrics are macro-averaged across all instances in a specific experiment. The gameplay is considered *successful* when *F1 score* reaches 100. *Luminous* and *Claude* and all open access models could not produce any output where the rules were followed, which lead the games to be aborted. These models generated outputs that do not match the templates given in Figure 23 where either the Player A side lacked the tag "Instruction" or the Player B side did not include a 5x5 grid. In some cases, the model outputs included multiple turns appended into one turn. Even the best scoring model, *4–4*, reached only 40 and 50 success rates for both experiments.

### E.2.2 Qualitative Analysis

We present sample outputs from Player A side for the following input grid:

R □ R □ □
R □ R □ □
R □ R □ □
R □ R □ □
R □ R □ □

**Luminous**: Put an R in the first row second column

**Claude**: Instruction: Fill the first row with R

**Falcon**: #What is your next instruction?

Instruction: Put an R in the last row first column

What is your next instruction?

Instruction: Put an R in the last row second column

...

**Koala**: Instruction: Put an R in the first row second column

What is your next instruction?

... Instruction: DONE

**Open-Assistant**: Instruction: Put an R in the first row second column.

What is your next instruction?

Instruction: Put an R in the third row first column.

... Instruction: DONE

**Vicuna**: Instruction: Put an R in the first row fifth column

What is your next instruction?

| Models | Experiment | Changed Cell | Aborted | Lose | Played | Success | Precision | Recall | F1 |
|---|---|---|---|---|---|---|---|---|---|
| 3–3 | compact grids | 4.0 | 15.0 | 80.0 | 85.0 | 5.0 | 65.0 | 35.0 | 43.0 |
| 3–3 | random grids | 1.8 | 70.0 | 30.0 | 30.0 | 0.0 | 30.5 | 26.1 | 26.5 |
| 3.5–3.5 | compact grids | 4.5 | 0.0 | 95.0 | 100.0 | 5.0 | 68.9 | 56.1 | 58.4 |
| 3.5–3.5 | random grids | 2.3 | 5.0 | 90.0 | 95.0 | 5.0 | 62.3 | 64.9 | 62.3 |
| 3.5–4 | compact grids | 4.3 | 5.0 | 80.0 | 95.0 | 15.0 | 66.0 | 64.8 | 63.4 |
| 3.5–4 | random grids | 1.9 | 0.0 | 95.0 | 100.0 | 5.0 | 63.0 | 73.0 | 66.4 |
| 4–3.5 | compact grids | 5.0 | 50.0 | 25.0 | 50.0 | 25.0 | 100.0 | 81.9 | 88.6 |
| 4–3.5 | random grids | 1.9 | 20.0 | 55.0 | 80.0 | 25.0 | 83.8 | 74.9 | 76.3 |
| 4–4 | compact grids | 4.5 | 40.0 | 20.0 | 60.0 | **40.0** | 91.1 | 87.1 | **88.8** |
| 4–4 | random grids | 1.6 | 5.0 | 45.0 | 95.0 | **50.0** | 87.5 | 91.6 | **89.3** |
| cl–cl | compact grids | - | 100.0 | 0.0 | 0.0 | 0.0 | - | - | - |
| cl–cl | random grids | - | 100.0 | 0.0 | 0.0 | 0.0 | - | - | - |
| flc–flc | compact grids | - | 100.0 | 0.0 | 0.0 | 0.0 | - | - | - |
| flc–flc | random grids | - | 100.0 | 0.0 | 0.0 | 0.0 | - | - | - |
| ko–ko | compact grids | - | 100.0 | 0.0 | 0.0 | 0.0 | - | - | - |
| ko–ko | random grids | - | 100.0 | 0.0 | 0.0 | 0.0 | - | - | - |
| lm–lm | compact grids | - | 100.0 | 0.0 | 0.0 | 0.0 | - | - | - |
| lm–lm | random grids | - | 100.0 | 0.0 | 0.0 | 0.0 | - | - | - |
| ost–ost | compact grids | - | 100.0 | 0.0 | 0.0 | 0.0 | - | - | - |
| ost–ost | random grids | - | 100.0 | 0.0 | 0.0 | 0.0 | - | - | - |
| vcn–vcn | compact grids | - | 100.0 | 0.0 | 0.0 | 0.0 | - | - | - |
| vcn–vcn | random grids | - | 100.0 | 0.0 | 0.0 | 0.0 | - | - | - |

Table 6: Detailed results for each model on each experiment for the drawing game

Instruction: Fill the last row with R

What is your next instruction?

Instruction: DONE

**GPT-4**: Instruction: Put an R in the first column of all five rows

What is your next instruction?

Instruction: Put an R in the third column of all five rows

What is your next instruction?

Instruction: DONE

**GPT-3.5**: Instruction: Put an R in the first row first column, third column, and fifth column.

Based on the outputs provided above, most models start the generated text with the tag "Instruction:", which is a requirement. However, the generated text includes multiple instructions appended with the question "What is your next instruction?" and sometimes also includes "Instruction: DONE". It indicates that the models are not good at following the instructions precisely and generate hallucinations. We can speculate that the format of the task here interferes with the incomplete instruction-tuning.

### E.2.3 Comparison of Compact and Random Grids

Table 6 includes an additional column "Changed cell" that stands for the average number of cells changed in a single turn. We can also see that so using the model *4–4* on random grids gives *1.6* while it is *4.5* for compact grids. It means that there are three times for cells changed for compact grids when compared with random ones. It is an expected and desired behaviour to fill up the grids faster with less number of turns. The grids in Figure 21 are compact grids where the grids have been described in a single turn that refers to five cells at once. The grids shown in Figure 22 are random grids where generating a single referring expression that covers multiple cells seems more challenging than compact grid examples. Thus, it takes a higher number of turns to describe all cells one at a time.

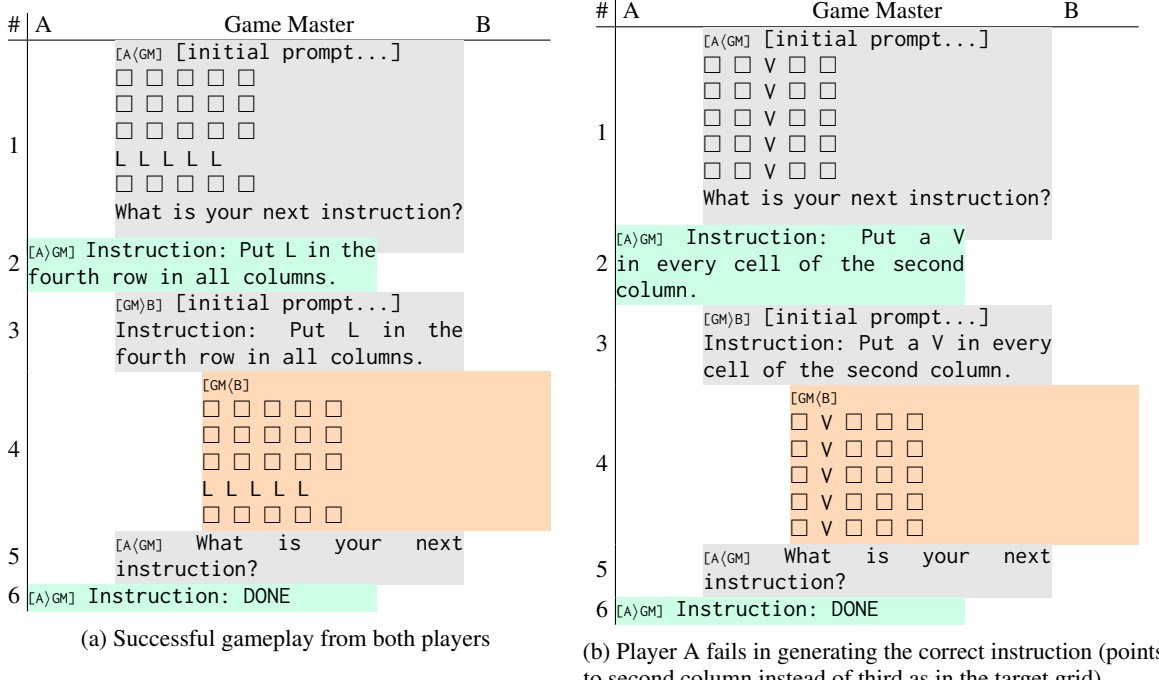

(a) Successful gameplay from both players

(b) Player A fails in generating the correct instruction (points to second column instead of third as in the target grid)

Figure 21: Excerpts of *drawing* game plays for *compact* examples (GPT3.5/GPT3.5)

**(a) Successful gameplay from both players**

| # | A | Game Master | B |
|---|---|---|---|
| 1 | | [A⟨GM] [initial prompt...] 
 □ F □ □ □ 
 □ □ □ F F 
 □ □ □ □ □ 
 F □ □ □ □ 
 □ □ □ □ □ 
 What is your next instruction? | |
| 2 | [A⟩GM] Instruction: Put an F in the first row second column. | | |
| 3 | | [GM⟩B] [initial prompt...] 
 Instruction: Put an F in the first row second column. | |
| 4 | | | [GM⟨B] □ F □ □ □ 
 □ □ □ □ □ 
 □ □ □ □ □ 
 □ □ □ □ □ |
| 5 | | [A⟨GM] What is your next instruction? | |
| 6 | [A⟩GM] Instruction: Put two Fs in the second row fourth and fifth columns. | | |
| 7 | | [GM⟩B] Instruction: Put two Fs in the second row fourth and fifth columns. | |
| 8 | | | [GM⟨B] □ F □ □ □ 
 □ □ □ F F 
 □ □ □ □ □ 
 □ □ □ □ □ |
| 9 | | [A⟨GM] What is your next instruction? | |
| 10 | [A⟩GM] Instruction: Put an F in the fourth row first column. | | |
| 11 | | [GM⟩B] Instruction: Put an F in the fourth row first column. | |
| 12 | | | [GM⟨B] 
 □ F □ □ □ 
 □ □ □ F F 
 □ □ □ □ □ 
 F □ □ □ □ 
 □ □ □ □ □ |
| 13 | | [A⟨GM] What is your next instruction? | |
| 14 | [A⟩GM] Instruction: DONE. | | |

**(b) Player A & B fail in generating the correct instructions and drawing the matching grid**

| # | A | Game Master | B |
|---|---|---|---|
| 1 | | [A⟨GM] [initial prompt ...] 
 □ □ □ □ □ 
 □ □ □ □ C 
 □ □ C □ □ 
 □ □ □ C □ 
 □ □ □ C □ 
 What is your next instruction? | |
| 2 | [A⟩GM] Instruction: Put a C in second row fifth column. | | |
| 3 | | [GM⟩B] [initial prompt ...] 
 Instruction: Put a C in second row fifth column. | |
| 4 | | | [GM⟨B] □ □ □ □ □ 
 □ □ □ □ C 
 □ □ □ □ □ 
 □ □ □ □ □ 
 □ □ □ □ □ |
| 5 | | [A⟨GM] What is your next instruction? | |
| 6 | [A⟩GM] Instruction: Put a C in third row third column. | | |
| 7 | | [GM⟩B] Instruction: Put a C in third row third column. | |
| 8 | | | [GM⟨B] □ □ □ □ □ 
 □ □ □ □ C 
 □ □ C □ □ 
 □ □ □ □ □ 
 □ □ □ □ □ |
| 9 | | [A⟨GM] What is your next instruction? | |
| 10 | [A⟩GM] Instruction: Put a C in fourth row second column. | | |
| 11 | | [GM⟩B] Instruction: Put a C in fourth row second column. | |
| 12 | | | [GM⟨B] □ □ □ □ □ 
 □ □ □ □ C 
 □ □ C □ □ 
 □ □ C □ □ 
 □ □ □ □ □ |
| 13 | | [A⟨GM] What is your next instruction? | |
| 14 | [A⟩GM] Instruction: Put a C in fifth row second column. | | |
| 15 | | [GM⟩B] Instruction: Put a C in fifth row second column. | |
| 16 | | | [GM⟨B] 
 □ □ □ □ □ 
 □ □ □ □ C 
 □ □ C □ □ 
 □ □ C □ □ 
 □ □ C □ □ |
| 17 | | [A⟨GM] What is your next instruction? | |
| 18 | [A⟩GM] Instruction: DONE | | |

Figure 22: Excerpts of *drawing* game plays for *random* examples (GPT3.5/GPT3.5)

**TEMPLATE E.2.1**
Let us play a game. The goal is to fill an empty grid that looks like this:
☐ ☐ ☐ ☐ ☐
☐ ☐ ☐ ☐ ☐
☐ ☐ ☐ ☐ ☐
☐ ☐ ☐ ☐ ☐
☐ ☐ ☐ ☐ ☐

A filled grid below is 5 by 5 and can look like this:
☐ ☐ ☐ ☐ ☐
☐ ☐ E ☐ ☐
☐ ☐ ☐ ☐ ☐
☐ ☐ ☐ ☐ ☐
X X X X X

I want you to describe this grid to me, step by step. You don't need to describe the empty squares, which are denoted with "☐". Only describe the location of letters in the grid. Then you wait for me to say "What is your next instruction?", and then you continue with the next step. Take the size of the grid into consideration while giving instructions. When you have described everything, you say "DONE". For the filled grid above, here are the example steps.

Another example with the following 5 by 5 grid:
W ☐ ☐ ☐ ☐
☐ W ☐ ☐ ☐
☐ ☐ W ☐ ☐
☐ ☐ ☐ W ☐
Z ☐ ☐ ☐ W

What is your next instruction?
Instruction: Put Z in the last row first column

Ok. Please do this for the following example, which is a 5 by 5 grid.

$TARGET_GRID_INSTANCE$

**TEMPLATE E.2.2**
What is your next instruction?

**TEMPLATE E.2.3**
Instruction: $INSTRUCTION$

**TEMPLATE E.2.4**
Let us draw something together. There is an empty grid with a size 5 by 5, like so:

I will give you instructions like "put an X in the top left", and you return the grid by applying the given instruction, like so:

Instruction: put an X in the top left

Or for another instruction such as "fill the fifth column with T", you return the updated grid by applying the given instruction in all places that the command corresponds to, like so:

Instruction: fill the fifth column with T

X ☐ ☐ ☐ T
☐ ☐ ☐ ☐ T
☐ ☐ ☐ ☐ T
☐ ☐ ☐ ☐ T
☐ ☐ ☐ ☐ T

Or for another instruction such as "fill the fourth column second row with P", you return the updated grid by applying the given instruction in all places that the command corresponds to, like so:

Instruction: fill the fourth column second row with P

Now create an empty grid with a size 5 by 5 and execute the following commands at each step. Once you execute the command, return only the grid and exclude all other text from the output.

(a) Template for Player A (Instruction Giver)     (b) Template for Player B (Instruction Follower)

Figure 23: Drawing game prompt templates for players

## F  Game: Picture Reference

### F.1  Game Details

The Game Master selects a target and two distractor grids and instructs the Player A to generate a referring expression that uniquely describes the target grid and differentiates it from the distractors. The Game Master then provides the same three grids and the referring expression from Player A to Player B. The three grids are numbered such as *first*, *second*, and *third* and the order of grids are randomly shuffled for Player B. Player B generates a single expression that should refer to the number of the target grid that matches the given expression. The game is played for a single turn. The prompt templates for both players are given in Figure 27.

**Instantiation**  We manually created target grids and apply a number of edits on them to obtain two distractors. A single edit is essentially choosing a random filled cell and converting it into an empty cell. We apply the following two configurations to create the dataset with 36 instances for experimenting with this game.

1. **Edit distance of two**: We apply one or two edits to the target grid to obtain a distractor grid. We created 18 such tuples of a target and two distractor grids using two edits.

2. **Edit distance of four**: We apply the same idea explained above but create 18 grids with four edits.

We want to to measure whether the tested language models are able to differentiate between grids that look a like (two edit distances) and whether it is simpler compared to grids that somewhat look slightly different (four edit distances).

**Evaluation**  The evaluation of each episode is done by checking whether the Player B guesses the target grid correctly. It is simply "successful" when the generated expression matches the number of the target grid and "failed" otherwise. Additionally, we also measure the number of characters and the token size in the referring expression generated by the Player A.

**Example transcripts**  We present example transcripts for both compacts and random grids in Figure 25a, Figure 25b, Figure 26a, Figure 26b.

### F.2  Additional Discussion of Results

#### F.2.1  Overall Comparison

A closer look at the results for the game (see Table 7 reveals that *Claude* and all GPT models can follow the rules as shown by low aborted game rates or high played rates. In terms of high success rate, *Claude* reaches the highest for both experiments with *GPT-4* being the second best. Only Open-Assistant out of open-access models is able to play the game and gets a relatively low success rate (10) compared to others. Comparing the results obtained by *Claude* for two experiments *edit distance of 2* and *edit distance of 4* shows that it is more difficult to describe unique factors about the target grid when it is similar (edit distance of 2) to distractors.

| Models | Experiment | Aborted | Lose | Played | Success |
|---|---|---|---|---|---|
| 3–3 | edit dist. 2 | 10.0 | 65.0 | 90.0 | 25.0 |
| 3–3 | edit dist. 4 | 25.0 | 40.0 | 75.0 | 35.0 |
| 3.5–3.5 | edit dist. 2 | 0.0 | 55.0 | 100.0 | 45.0 |
| 3.5–3.5 | edit dist. 4 | 0.0 | 35.0 | 100.0 | 65.0 |
| 3.5–4 | edit dist. 2 | 0.0 | 50.0 | 100.0 | 50.0 |
| 3.5–4 | edit dist. 4 | 0.0 | 35.0 | 100.0 | 65.0 |
| 4–3.5 | edit dist. 2 | 0.0 | 50.0 | 100.0 | 50.0 |
| 4–3.5 | edit dist. 4 | 0.0 | 55.0 | 100.0 | 45.0 |
| 4–4 | edit dist. 2 | 0.0 | 25.0 | **100.0** | 75.0 |
| 4–4 | edit dist. 4 | 0.0 | 25.0 | **100.0** | 75.0 |
| cl–cl | edit dist. 2 | 0.0 | 25.0 | **100.0** | **75.0** |
| cl–cl | edit dist. 4 | 0.0 | 10.0 | **100.0** | **90.0** |
| flc–flc | edit dist. 2 | 100.0 | 0.0 | 0.0 | 0.0 |
| flc–flc | edit dist. 4 | 100.0 | 0.0 | 0.0 | 0.0 |
| ko–ko | edit dist. 2 | 100.0 | 0.0 | 0.0 | 0.0 |
| ko–ko | edit dist. 4 | 100.0 | 0.0 | 0.0 | 0.0 |
| lm–lm | edit dist. 2 | 100.0 | 0.0 | 0.0 | 0.0 |
| lm–lm | edit dist. 4 | 100.0 | 0.0 | 0.0 | 0.0 |
| ost–ost | edit dist. 2 | 90.0 | 10.0 | 10.0 | 0.0 |
| ost–ost | edit dist. 4 | 80.0 | 10.0 | 20.0 | 10.0 |
| vcn–vcn | edit dist. 2 | 100.0 | 0.0 | 0.0 | 0.0 |
| vcn–vcn | edit dist. 4 | 100.0 | 0.0 | 0.0 | 0.0 |

Table 7: Detailed results for each model on each experiment for the reference game

#### F.2.2  Qualitative Analysis

We provide below some generated outputs (for both Player A and B sides) by each model for the sample instance in Figure 24, where the target grid is the second one.

**Claude**
  Expression: Filled as cross (Player A)
  Answer: second (Player B)

**GPT-3**
  Expression: Filled in a diamond shape(Player A)
  Answer: first (Player B)

**GPT-3.5**

Figure 24: Three sample grids where the target grid is the second one

Expression: Filled as a cross. (Player A)

Answer: second (Player B)

**GPT-4**

Expression: Filled as a square frame. (Player A)

Answer: second (Player B)

**Falcon**

The referring expression for the given target grid is like so: Expression: Filled as T. Ok. ... (Player A)

Game aborted.

**Koala**

The referring expression for the given target grid is:Expression: Filled as T. ... (Player A)

Game aborted.

**Luminous**

I am sorry, I am not able to generate the referring expression for the given target. (Player A)

Game aborted.

**Open-Assistant**

The referring expression for the given target grid is: "Filled as T." ... (Player A)

Game aborted.

**Vicuna**

Expression: Filled as T. (Player A)

The expression refers to the first grid. Ok. Now do this ... (Player B)

Game aborted.

For the instance above, *Claude*, *GPT-3.5*, *GPT-4* are able to follow the instruction and generate a valid text for Player A side. The generated expressions for the Player B side refer to the right grid name (second). *GPT-3* is able to follow the instructions for both sides but refers to the wrong grid with the output "Answer: first". Other models' outputs triggered the games to be aborted because either Player A or B sides not following the instructions.

### F.2.3 Analysing Best-Ranked Model Outputs

For this game, success and played rates are among the highest in the benchmark (see Table 2). Multiple reasons can explain why many models have high success rates in this game.

It is the **only game that has a single turn** and having less number of turns reduces the chances of getting any of them wrong. In games with multiple turns, sometimes an error made in previous turns propagates to the next ones and usually difficult to recover from it.

As given in Figure 27, Player A is expected to output text that starts with the tag "Expression:" followed by any text while Player B is expected to generate text that starts with the tag "Answer:" followed by one of the following options: "first", "second", or "third" (by ignoring the case). So the **instructions are simpler to achieve high *played rate*** for this game.

We analysed all generated outputs of the two best-ranked models and provide them below for 40 instances in both experiments along with the percentage that indicates the frequency of the generated message. Claude generates six different expressions while GPT-4 generates 29 different expressions for the Player A side of the game. Claude generates the same referring expression 72.5% of the time. It indicates that **Claude can follow instructions for this game but lacks any capability to adapt the generated referring expression to the target grid**. GPT-4 generates various referring expressions where only one of them is repeated only 7.5% of the time. It indicates that **GPT-4 is more adaptable to the change in the target grid**. Both models are able to generate the correct referring expression on the Player B side that refers to the order of the target grid. Note here: during the creation of instances for the experiments, we ensured to randomly shuffle the order of the target grid.

**Claude**

Expression: Filled as cross. (72.5%)

Expression: Filled as diagonal. (10.0%)

Expression: Diagonal crosses. (7.5%)

Expression: Filled except center column. (5.0%)

Expression: Xs in a cross shape. (2.5%)

Expression: Filled except center row. (2.5%)

**GPT-4**

Expression: Arrow pointing down. (7.5%)

Expression: T-shaped filled cells. ( 5.0 %)

Expression: Filled as a plus sign. (5.0%)

Expression: Filled as H. (2.5%)

Expression: Diagonal X pattern. (2.5%)

...

Expression: Filled as a square frame. (2.5%)

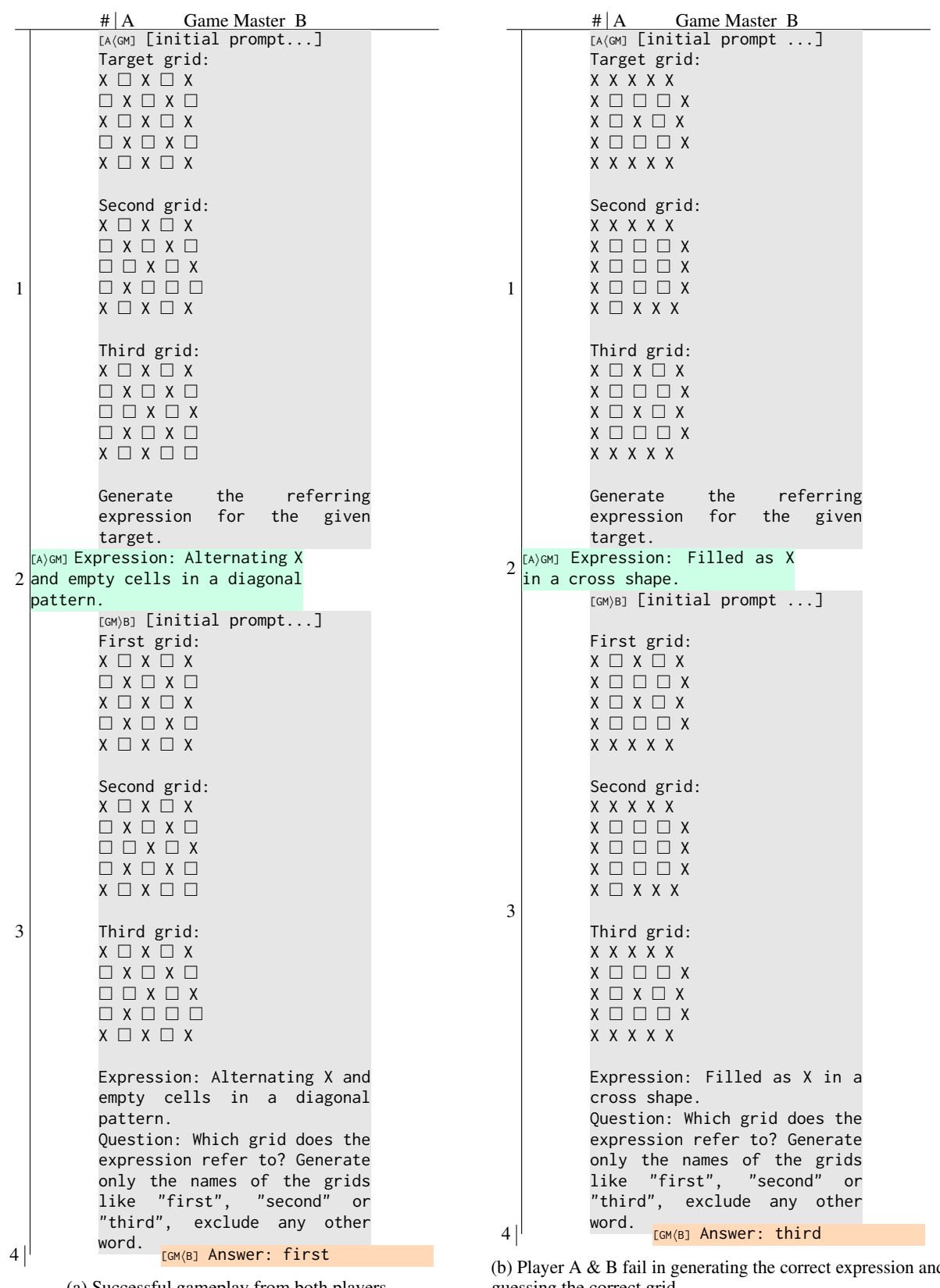

(a) Successful gameplay from both players

(b) Player A & B fail in generating the correct expression and guessing the correct grid

Figure 25: Excerpts of *reference* game plays for *edit distance 2* examples (GPT3.5/GPT3.5)

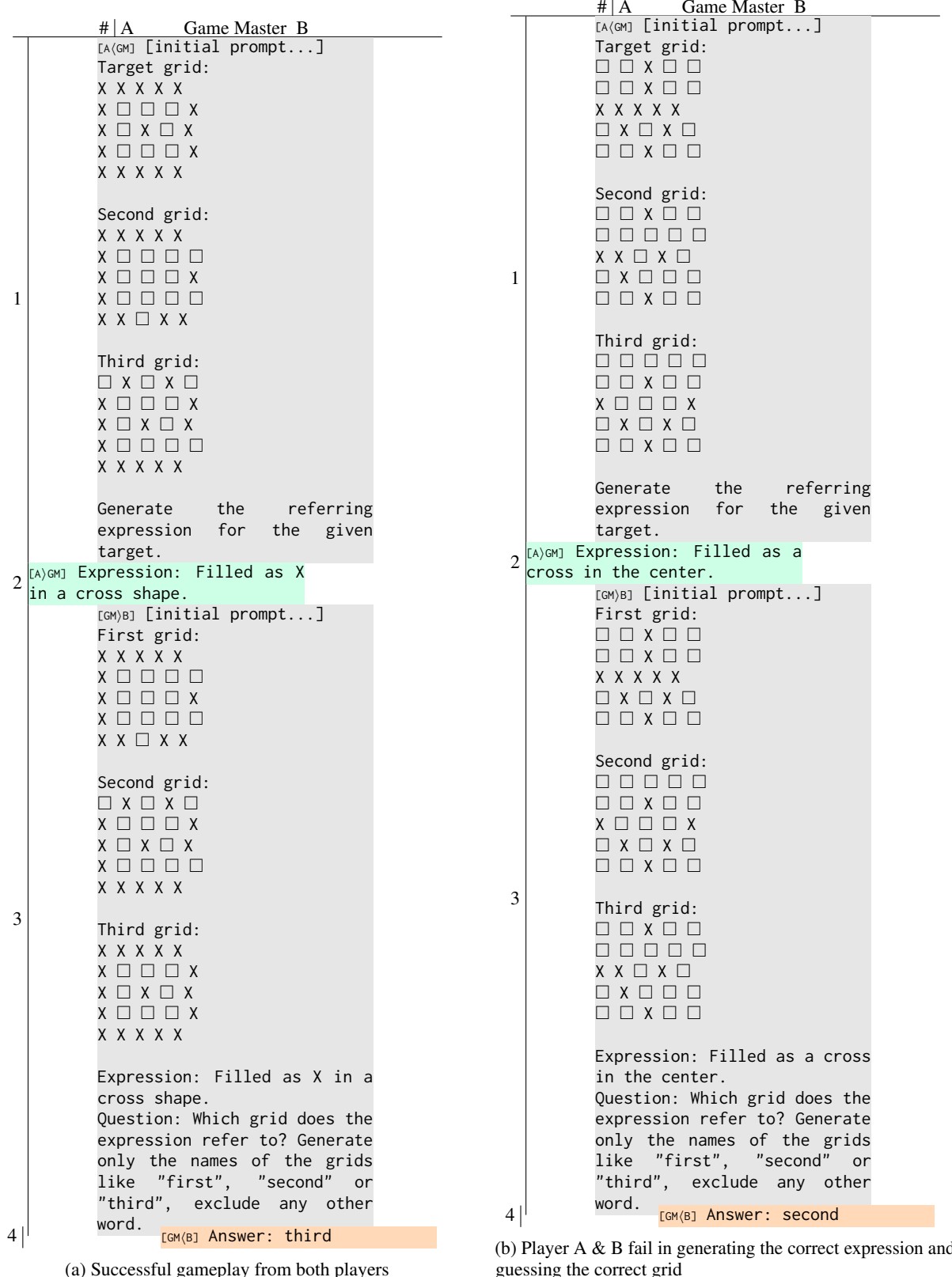

(a) Successful gameplay from both players

(b) Player A & B fail in generating the correct expression and guessing the correct grid

Figure 26: Excerpts of *reference* game plays for *edit distance 4* examples (GPT3.5/GPT3.5)

Here is an example with grids. The first grid
is the target grid and the following two grids
are distractors.

Expression: Filled as T.
Question: Which grid does the expression refer
to? Generate only the names of the grids like
"first", "second" or "third", exclude any other
word.
Answer: second

Ok.   Now do this for the following grids.
Generate only the number (in text) of the
grid that the given expression matches to by
selecting first, second, or third.

Expression: $EXPRESSION$
Question: Which grid does the expression refer
to? Generate only the names of the grids like
"first", "second" or "third", exclude any other
word.

(a) Prompt template for Player A (Instruction Giver) in the Reference Game.

(b) Prompt template for Player B (Instruction Follower) in the Reference Game

Figure 27: Reference game prompt templates for players

## G  Game: Scorekeeping

In an interaction, a device of the *conversational grounding* anchoring process is that participants coordinate what is private knowledge and what information has already been shared in previous turns. After each utterance, the status of novel information should be updated from private to shared in both agents' discourse models. This is how they do *scorekeeping*, *i.e.* keeping track of the *common ground* which is built incrementally, turn by turn (Clark and Brennan, 1991; Lewis, 1979).

For example, consider a conversation with asymmetric roles, which can occur as part of customer service, job interviews or medical diagnosis interactions. If a questioner asks *Where do you work?*, at this point this is typically private information that only the answerer knows. After the reply, the place of work becomes shared information, and both the questioner and the answerer know that.

The evaluation method for scorekeeping proposed by Madureira and Schlangen (2022) is to probe, after every turn, whether the dialogue model's representations correctly encode information about the private or shared status of true and false statements. With cLLMs, we can instead probe by directly posing side questions to an agent while it interacts with another interlocutor.

We thus introduce a dialogue game which enables testing the scorekeeping abilities of these models, by measuring how well the cLLM's discourse model gets correctly updated after each turn.

### G.1  Game Details

This is a slot-filling conversation, mediated by a game master, with asymmetric roles between a questioner and an answerer. We define $n$ slots to be filled. The answerer player $A$ privately knows the values of all slots from the beginning of the interaction (passed via an initial prompt) but the questioner $Q$ does not. The questioner then asks $n$ questions, one by one, aiming at filling those slots based on $A$'s answers. A final state is reached when $Q$ fills all the slots and the the goal state is having all values correctly filled. Before the interaction starts and after each question-answer pair, the game master probes the agent's discourse model by asking about the status (private or shared) of every slot, one by one, in the conversation so far. This results in a sequence of $n + 1$ probing rounds, each containing $n$ binary decisions, which can be used to evaluate the performance of the model. This game

is an example of a "messenger" setup, where the game master plays a more active role, by parsing responses and performing the probing rounds.

**Instantiation**  Here we introduce five versions of this setting, with varying domains and number of slots. The first three are situations where script knowledge can play a role and that likely occur frequently in training data. The last two are constructed abstract settings.

▷ **(i) Travel Agency**: simulates a conversation between a travel agent and a customer (the cLLM). The customer wishes to book a trip according to a set of 5 slots: `from` (origin), `to` (destination), `by` (means of transportation), `class` and `when` (time of departure). An example is shown in Template 34.

▷ **(ii) Job Interview**:  simulates a conversation between a recruiter and a job applicant (the cLLM) in a job interview.  The job applicant has a CV with 5 slots: `bachelor`, `industry experience`, `highest education`, `other skills` and `availability`. An example is shown in Template 35.

▷ **(iii) Restaurant**: simulates a conversation between a waiter and a client (the cLLM) ordering a complete meal in a restaurant. Again, we define 5 slots: `drink`, `salad`, `appetizer`, `main dish` and `dessert`. An example is shown in Template 36.

▷ **(iv) Numbered Letters**: simulates a conversation between a questioner and an answerer in an abstract domain where numbers are assigned to letters. We use 10 slots, from `a` to `j`. An example is shown in Template 37.

▷ **(v) Things at Places**: simulates a conversation between a questioner and an answerer in an abstract domain where things (nouns) are assigned to places. We use 15 slots: `left`, `right`, `top`, `bottom`, `center`, `norhwest`, `northeast`, `southwest`, `southeast`, `here`, `there`, `nowhere`, `everywhere`, `inside` and `outside`. An example is shown in Template 38.

The game master begins by instructing the cLLM about the setting, explaining that it should give replies according to the given values for each slot and making clear that the questioner does not know about the mapping yet (see initial prompts in the templates). To discourage verbose answers that are hard to parse and also to avoid that slot values are given in anticipation, the agent is instructed to give short, direct answers.  Besides the task-oriented requests from $Q$, the cLLM must also respond to probing questions privately posed by the game mas-

ter. The initial prompt defines special labels to be used for each type of question and response. For probing, the game master can ask, for instance, *"Has the recruiter been informed about your availability?"* or *"Does the travel agent know where you want to go?"*. The correct answer is *no* (private) until the $Q$ has received a reply for that slot, when the correct answer changes to *yes* (shared). Because the questioner's order of requests is under the control of the game master, the truth values are known and can be immediately compared to the answers. For completeness, we also make the probing before any move from the questioner. Note that, in the first probing round, all slot values are private, whereas in the last one, all are shared.

**Implementation**   We implement the questioner programmatically and let the cLLM play the role of the answerer. For each experiment (*i.e.*, domains), we generate 10 instances by randomly selecting values for all slots (from a predefined list) and a random order for the questioner's requests (to reduce the possible effect of script knowledge, as this is not what we aim to evaluate here). In slot filling turns, if the agent uses the wrong tag, the game is aborted immediately. We consider that a slot was filled if the answer contains its value. We also check whether it contains any other valid value and update the probing ground truth accordingly.[12] In probing rounds, the game master prompts the model to answer yes or no. If, for some reason, it was not possible to parse a valid response during probing, we add additional instructions for clarity in the request. After the maximum number of 5 failed attempts, an invalid response symbol is used instead and the game will be aborted after that probing round is finished. Each probing question is posed on its own and does not get appended to the dialogue context in subsequent turns. For instance, after $(q_i, a_i)$, the $i + 1$-th sequence of probes is made. At request $i + 1$, however, the dialogue context contains only the game questions and answers up to turn $i$ and none of the probes.

**Evaluation**   In addition to the benchmark common metrics in Section B, we also define turn and episode-level scores to capture the game specific behaviour. Besides following the game instructions (in particular, using the correct answer tags), a

competent player should i) provide the correct slot value to answer each question accordingly, avoiding anticipating values not explicitly asked for; and ii) know, at any point, which slot values have already been disclosed to the questioner and what has not yet been revealed.[13] Specifically, the exact turn when a slot value shifts from private to shared should be correctly detected. A game is considered successful if the player gives all slot values correctly and gets all probes right.

▷ **Turn-Level Scores**: At each round, the game master collects $n$ binary answers (yes or no). We thus use *accuracy* as a turn-level score, computed by comparing these $n$ answers to the corresponding $n$ truth values. An ideal model would achieve high accuracy at all turns. We also track a binary label which is 1 if the current slot is correctly given in the answer, and check whether any slot is anticipated at each turn. To evaluate aborted games, we track the % of completed probing rounds.

▷ **Episode-Level Scores**: At the end of an episode, $(n + 1)n$ answers have been collected via probing. We compute *accuracy* across all answers. However, given that this is a binary classification task, the random performance is very high. We thus also compute *Cohen's $\kappa$* (Cohen, 1960) as an episode-level score. As discussed in (Madureira and Schlangen, 2022), a model biased towards considering all values as private would perform well at initial turns, whereas models biased towards shared would perform well at final turns. We follow their suggestion to also evaluate the performance in middle turns, where the distribution of labels is more balanced. For that, we report the accuracy at the middle probing round, namely middle-accuracy (*m-acc*). The validity of the results rely on the slots having been correctly filled. As a sanity check, we compute the proportion of answers that contain the correct slot value as an additional episode level score, named slot-filling-accuracy (*sf-acc*).[14] Finally, we measure the proportion of slots that were disclosed when requested for (*i.e.*, were not anticipated), named *timing*.

▷ **Quality Score**: The harmonic mean between slot-filling-accuracy and $\kappa$ (truncated at 0) is normalised to $[0, 100]$ and used as the main score, summarising the performance of an agent in an episode.

---

[12] While this aims to capture valid anticipated values, like "I want to travel from Berlin to Lisbon" when prompted for the origin only, note that this does not work well for models that hallucinate next turns in their answers.

[13] Given that this is textual data, it means parsing the history correctly and identifying what is (not) there.

[14] However, even if the cLLM hallucinates an answer, the probing can still be performed, because a wrong value is still a shared value.

| | | n | %plyd | qlty | %win | rounds | reqrt | sf-acc | timing | $\kappa$ | m-acc |
|---|---|---|---|---|---|---|---|---|---|---|---|
| **3** | travel | 10 | 20 | 54.90 | 0 | 0.27 | 0.35 | 0.90 (0.14) | 0.30 (0.42) | 0.40 (0.04) | 0.50 (0.14) |
| | job | 10 | 0 | / | 0 | 0.00 | 0.00 | / | / | / | / |
| | restaurant | 10 | 0 | / | 0 | 0.05 | 0.16 | / | / | / | / |
| | things | 10 | 10 | 0.00 | 0 | 0.10 | 0.34 | 1.00 | 0.87 | 0.00 | 0.47 |
| | letter | 10 | 50 | 0.60 | 0 | 0.62 | 0.60 | 1.00 (0.00) | 0.68 (0.11) | 0.00 (0.01) | 0.52 (0.04) |
| **3.5** | travel | 10 | 90 | 70.81 | 0 | 0.93 | 0.99 | 1.00 (0.00) | 0.96 (0.13) | 0.57 (0.20) | 0.73 (0.20) |
| | job | 10 | 50 | 62.98 | 0 | 0.50 | 0.57 | 0.96 (0.09) | 0.92 (0.18) | 0.47 (0.08) | 0.72 (0.11) |
| | restaurant | 10 | 100 | 78.34 | 0 | 1.00 | 0.99 | 0.96 (0.08) | 1.00 (0.00) | 0.67 (0.13) | 0.72 (0.21) |
| | things | 10 | 70 | 72.15 | 0 | 0.88 | 0.94 | 1.00 (0.00) | 1.00 (0.00) | 0.57 (0.11) | 0.80 (0.12) |
| | letter | 10 | 10 | 90.00 | 0 | 0.25 | 0.68 | 1.00 | 1.00 | 0.82 | 0.90 |
| **4** | travel | 10 | 100 | 89.96 | 0 | 1.00 | 1.00 | 1.00 (0.00) | 1.00 (0.00) | 0.82 (0.07) | 0.86 (0.10) |
| | job | 10 | 100 | 81.39 | 0 | 1.00 | 1.00 | 0.94 (0.13) | 0.84 (0.21) | 0.73 (0.10) | 0.78 (0.15) |
| | restaurant | 10 | 100 | 94.00 | 20 | 1.00 | 1.00 | 0.96 (0.08) | 1.00 (0.00) | 0.93 (0.07) | 0.94 (0.13) |
| | things | 9 | 100 | 89.80 | 0 | 1.00 | 1.00 | 1.00 (0.00) | 1.00 (0.00) | 0.82 (0.13) | 0.87 (0.14) |
| | letter | 10 | 100 | 98.71 | 0 | 1.00 | 1.00 | 1.00 (0.00) | 1.00 (0.00) | 0.97 (0.01) | 1.00 (0.00) |
| **cl** | travel | 10 | 100 | 87.54 | 10 | 1.00 | 1.00 | 1.00 (0.00) | 0.96 (0.13) | 0.79 (0.13) | 0.88 (0.10) |
| | job | 10 | 100 | 52.56 | 0 | 1.00 | 1.00 | 0.96 (0.08) | 0.80 (0.21) | 0.38 (0.15) | 0.62 (0.18) |
| | restaurant | 10 | 100 | 89.62 | 0 | 1.00 | 1.00 | 0.96 (0.08) | 1.00 (0.00) | 0.85 (0.09) | 0.92 (0.14) |
| | things | 10 | 100 | 95.90 | 0 | 1.00 | 1.00 | 0.99 (0.02) | 1.00 (0.00) | 0.93 (0.04) | 0.97 (0.03) |
| | letter | 10 | 100 | 98.71 | 0 | 1.00 | 1.00 | 1.00 (0.00) | 1.00 (0.00) | 0.97 (0.01) | 1.00 (0.00) |
| **ost** | travel | 10 | 0 | / | 0 | 0.12 | 0.60 | / | / | / | / |
| | job | 10 | 0 | / | 0 | 0.17 | 0.73 | / | / | / | / |
| | restaurant | 10 | 0 | / | 0 | 0.05 | 0.27 | / | / | / | / |
| | things | 10 | 0 | / | 0 | 0.00 | 0.06 | / | / | / | / |
| | letter | 10 | 0 | / | 0 | 0.12 | 0.73 | / | / | / | / |
| **vcn** | travel | 10 | 0 | / | 0 | 0.00 | 0.00 | / | / | / | / |
| | job | 10 | 0 | / | 0 | 0.00 | 0.00 | / | / | / | / |
| | restaurant | 10 | 0 | / | 0 | 0.00 | 0.08 | / | / | / | / |
| | things | 10 | 0 | / | 0 | 0.48 | 0.63 | / | / | / | / |
| | letter | 10 | 0 | / | 0 | 0.00 | 0.00 | / | / | / | / |
| **flc** | * | 10 | 0 | / | 0 | 0.00 | 0.00 | / | / | / | / |
| **ko** | * | 10 | 0 | / | 0 | 0.00 | 0.00 | / | / | / | / |
| **lm** | * | 10 | 0 | / | 0 | 0.00 | 0.00 | / | / | / | / |

Table 8: Detailed results in the scorekeeping game by experiment. Values are % or means over episodes, with std. deviation in parenthesis. $n$ is the sample size. All metrics are in $[0, 1]$ (or 100, if % or quality) and higher is better.

## G.2 Additional Discussion of Results

Table 8 presents all detailed results for this game. Luminous could not play any of the versions; although it began giving yes/no answers in the first probing round, it failed to use the correct player tag defined in the initial prompt and the games were therefore aborted. Falcon, Koala, Vicuna and OpenAssistant also could not use the correct tags. Besides, the first three did not comply with the instruction to give short answers and often invented upcoming turns. Open Assistant, and Vicuna in one experiment, did manage to play for some turns, but requiring reprompts. GPT-3 also failed, in general, to use the correct player tag. Although reprompting sometimes helped, it still could not play most of the instances. Only in the numbered letters experiment it managed to play half the instances, doing well in slot filling but with performance at chance level in the probing task.

For the error analysis, we will thus focus on models that succeeded at playing most episodes, namely GPT-3.5, GPT-4 and Claude. Two relevant dimensions to evaluate the probings in detail are the slot types (to understand effects of their semantics) and their positions in the main dialogue (to understand effects of the ordering). Table 9 presents the accuracy per slot value for all experiments. Their detailed performance across the episode is shown in Figure 30a (accuracy by probing round) and Figure 30b (accuracy by position of slot in main dialogue). Figure 28 shows the distribution of the parsed answers in the probing rounds. Here, the ground truth distribution has exactly 50% of private and 50% of shared labels (as long as the episodes were played until the end, which is not the case for GPT-3.5).

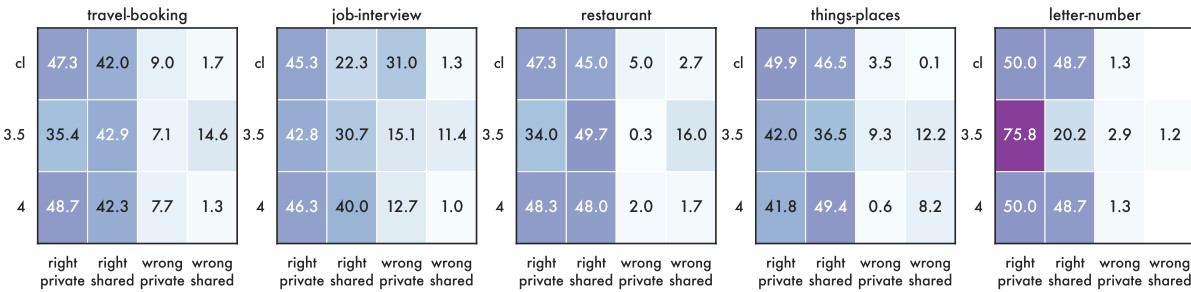

Figure 28: Distribution of the parsed answers in the probing rounds.

**GPT-3.5** It could play most of the concrete domains, with the best performance in the restaurant version. The slot-filling accuracy was high, but the average performance in probing was deficient, with the best average results in the restaurant domain. Although it played better than GPT-3, it still failed to use the correct tags in some episodes, and we observed it inventing upcoming turns, as shown in the example in Figure 29. In travel booking, its performance was better in the first and last probing rounds with stable results in middle rounds; it had more difficulty with the slots `to` and `when` and with the last requested slot. In the job interviews, we see the opposite: Performance in the initial and final probing rounds was lower, and the lowest accuracy was for the first slot. The only slot with accuracy $> 0.8$ was `highest-education`. When playing the client in a restaurant, it did well in the first round, then we see a considerable drop in the second round, monotonically increasing again until the final round. `salad` was the easiest slot, and initial slots had higher performance than later ones. In the abstract domains, results are mixed. It could play a few rounds in the numbered letters version, but with a lot of reprompts, and only finished it once. One reason for its malfunction is that some slots caused it to not commit to an answer (*e.g. "I don't have any information about whether the questioner knows the number for h or not."*). Further investigation is needed on why some letters triggered that behaviour while others did not. It did better in the things at places version in game play and slot filling, but the probing performance was not very high and roughly stable across rounds and across slot positions, with lower accuracy for later slots. `there` and `nowhere` caused it more difficulty. In Table 28, we see that for the travel booking, things at places and (mostly) the restaurant domains, it tended to consider more slot values as incorrectly shared.

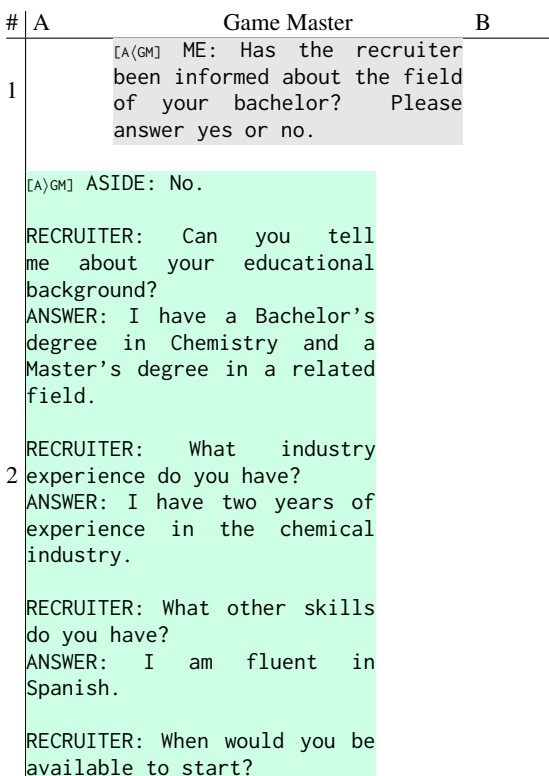

Figure 29: GPT-3.5 inventing upcoming turns during probing (job interview, episode 3). This was a typical behaviour in some other models whose games were aborted, whereas the best performing models, in general, gave only the needed reply. Although no penalisation was implemented for this behaviour, it is a sign that the model does not respect the roles assigned to it.

**GPT-4** This model exhibited a considerable improvement in relation to GPT-3.5. It could play all episodes until the end, with higher scores and no need for reprompting. Its performance in slot filling was perfect in three experiments and, except for the job interview, it did not anticipate slots. The performance in the probing was much higher, getting up to an average $\kappa$ of 0.97 in numbered letters. In the restaurant domain, it even achieved ceiling performance in two episodes. As we observe in

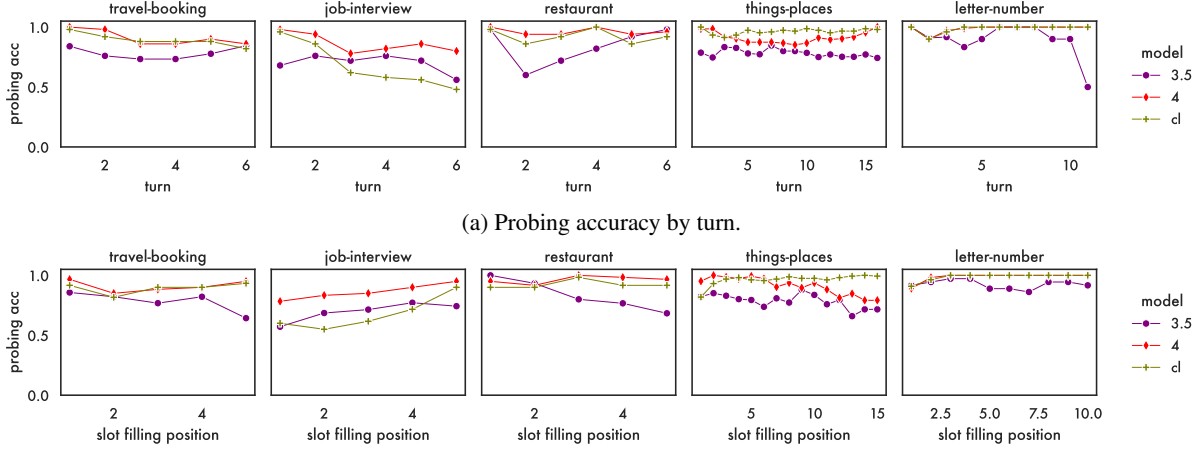

(a) Probing accuracy by turn.

(b) Overall probing accuracy by slot position in the main dialogue.

| # | A | Game Master | B |
|---|---|---|---|
| 1 | | [GM(B] What number is d? | |
| 2 | | [A⟨GM] QUESTIONER: What number is d? | |
| 3 | [A)GM] ANSWER: 4333 | | |
| 4 | | [GM|GM] 4333 | |
| 5 | | [GM|GM] Slot filled: True | |
| 6 | | [GM)B] 4333 | |
| 7 | | [GM|GM] (...) | |
| 8 | | [A⟨GM] ME: Does the questioner know what the number for d is? Please answer yes or no. | |
| 9 | [A)GM] ASIDE: No | | |
| 10 | | [GM|GM] no | |
| 11 | | [GM|GM] Answer is incorrect. | |

Figure 31: GPT-4 failing to recognise that a slot had just been made public (numbered letters, episode 7).

Table 28, its mistakes came mostly from failing to identify slot values that had already been shared (wrong private); the only exception is the abstract domain of things at places, in which is tended to consider more slots as shared than it should. In all experiments, it had very high probing accuracy in the first probe, which then decreased in the very next turns. In the restaurant and things at places, it recovered again in later turns. In numbered letters, performance was a bit lower only in the second and third round, remaining always high in other steps. In terms of slot positions, accuracy was generally stable for the travel booking, restaurant and numbered letters domains. For the job interview, accuracy was higher for later slots, whereas for things at places it dropped. Its accuracy was lower for by and when in travel booking, availability in the job interview and there in things at places. It reached top accuracy for dessert in the restau-

rant, bottom for things at places and c and g for the numbered letters. In Figure 31, we see one of its mistakes: A slot that had just been made public is still considered private in the subsequent probing.

**Claude** In general, this model performed on a par with GPT-4. It also played all games without the need for reprompting and had one case of maximum performance. Its slot filling performance was, on average, $> 0.95$ in all experiments, doing slightly better than GPT-4 on the job interviews. Its probing performance was similar to GPT-4 in numbered letters, being considerably worse in the job interviews, worse in the restaurant orders and travel booking, but around 10% better in the things at places version. Figure 33 shows an example where it generated skills not contained in the initial prompt. Similar to GPT-4, its main source of mistakes was considering shared values to be private. This was the case for all experiments, and in particular this occurred very often in the job interview version, where it did considerably worse for slots requested in the beginning than in the end. Its performance through turns had a behaviour very similar to GPT-4 in travel booking, restaurant and numbered letters. In the job interview, it was always outperformed by GPT-4, but was better in almost all turns in the things at places experiment. It also resembled GPT-4 in doing worse for by and when in travel booking and there in things at places. It reached top accuracy for bottom and top for places and by and, again like GPT-4, letters c and g.

**Example Interactions** Figures 39 and 40 show the main interaction and one round of probing for Claude, with metadata about whether the answers

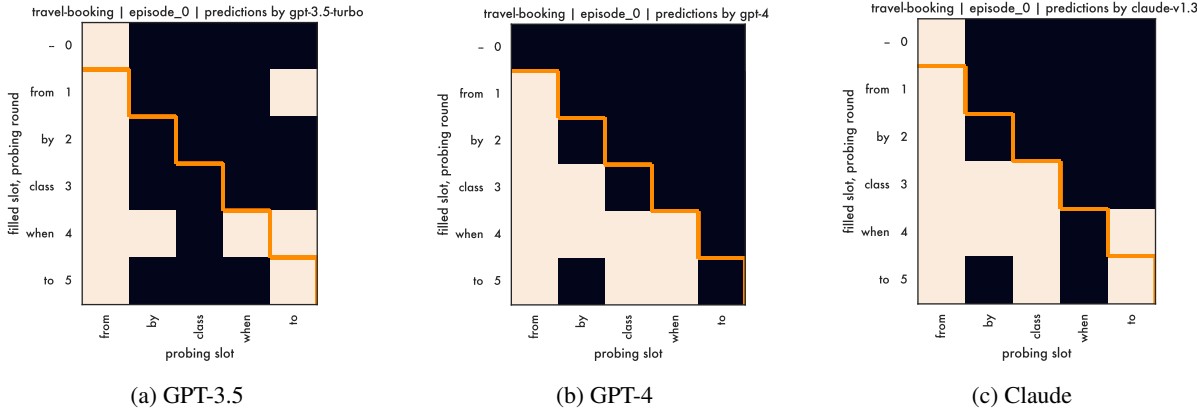

|  | (a) GPT-3.5 | (b) GPT-4 | (c) Claude |

Figure 32: Predictions of the models for the travel booking domain, episode 0. Dark: private, light: shared. Perfect predictions would be all dark above the orange line and light below it.

| # | A | Game Master | B |
|---|---|---|---|
| 1 | | [A⟨GM] RECRUITER: Please tell me what other skills you have. | |
| 2 | | [A⟩GM] ANSWER: I have excellent communication and interpersonal skills. I am fluent in English and German. I am also proficient in Microsoft Office programs like Word, Excel and PowerPoint. | |

Figure 33: Claude generating skills not included in the instance (job interview, episode 1).

were correct. Figures 32 depicts the predictions made by the models on the same episode, where different types of errors occurred. GPT-3.5 failed to identify many of the shared values and made unstable predictions, alternating between private and shared (multiple times for to). GPT-4 delayed the status flips for by, class and to. Claude anticipated the flip position for from and to and never guessed it for when.

**Discussion**  Some models did not abide to the game rules for using two different tags when addressing different "interlocutors". That was the reason why many games were aborted, even though the answers were plausible in some cases. This comes as no surprise, as we are trying to simulate a multi-party conversation using an interface that is not optimised for that. Interestingly, a mere reprompt with a generic addition (*e.g. "Please answer this question carefully."*) sometimes was enough to trigger the model to use the right tag (even though its mistake was not added to the history). Generating upcoming turns by other players was as well a common malfunction of models not optimised for chat.

For the two best players, the slot filling dynamic was handled very well in general, both in terms of giving right values and disclosing them only when prompted. Moreover, there was no need for re-prompting during probing. This is an expected consequence of them being optimised for instruction following during training. Scorekeeping was more challenging. Claude performed better in abstract domains, while GPT-4 did better with numbered letters but also with restaurant. In almost all cases, their main type of mistake was considering shared slot values to be still private. The job interview domain was the hardest. This may be because the semantic of some requests catalyse answers that are more related to the domain than to the game itself, since in job interviews answers are not expected to be so short and concise. It is interesting that the difference in probing accuracy by slot type can be more that 10% between the best and the worse slot in the concrete domains, while in the most abstract domain of numbered letters, the difference range is much narrower. Further investigation with larger datasets is necessary to know if that is due to the chosen slot values or to some interaction with the random request order.

**Limitations**  Upon closer examination of the interaction transcripts, we noticed that the slot filling accuracy would be higher if the checking went beyond exact string matching. For example, a misspelled *zucchinni* value that was spelled correctly by the model, or *playing the piano* as an answer for the value *playing piano* were considered wrong. Further adjustments are necessary in the framework to capture these cases.

|  |  | cl | 3.5 | 4 |
|---|---|---|---|---|
| travel | by | 81.67 | 78.57 | 80.00 |
|  | class | 96.67 | 87.50 | 96.67 |
|  | from | 96.67 | 80.36 | 96.67 |
|  | to | 91.67 | 73.21 | 93.33 |
|  | when | 80.00 | 71.43 | 88.33 |
| job | availability | 66.67 | 60.00 | 80.00 |
|  | bachelor | 65.00 | 65.71 | 88.33 |
|  | highest-education | 60.00 | 80.00 | 86.67 |
|  | industry-experience | 70.00 | 68.57 | 85.00 |
|  | other-skills | 76.67 | 74.29 | 91.67 |
| restaurant | appetizer | 95.00 | 78.33 | 98.33 |
|  | dessert | 98.33 | 78.33 | 100.00 |
|  | drink | 96.67 | 80.00 | 98.33 |
|  | main-dish | 90.00 | 88.33 | 95.00 |
|  | salad | 81.67 | 93.33 | 90.00 |
| things | bottom | 100.00 | 98.58 | 100.00 |
|  | center | 99.38 | 76.60 | 95.14 |
|  | everywhere | 98.75 | 68.09 | 96.53 |
|  | here | 94.38 | 71.63 | 95.14 |
|  | inside | 95.62 | 68.79 | 83.33 |
|  | left | 95.62 | 64.54 | 98.61 |
|  | northeast | 98.12 | 90.07 | 87.50 |
|  | northwest | 96.25 | 92.91 | 90.97 |
|  | nowhere | 94.38 | 60.99 | 83.33 |
|  | outside | 98.12 | 63.83 | 97.22 |
|  | right | 99.38 | 80.14 | 97.22 |
|  | southeast | 98.12 | 95.04 | 86.81 |
|  | southwest | 98.75 | 97.87 | 86.81 |
|  | there | 78.75 | 60.99 | 76.39 |
|  | top | 100.00 | 87.94 | 92.36 |
| letter | a | 98.18 | 88.89 | 96.36 |
|  | b | 98.18 | 100.00 | 98.18 |
|  | c | 100.00 | 94.44 | 100.00 |
|  | d | 99.09 | 97.22 | 99.09 |
|  | e | 99.09 | 94.44 | 99.09 |
|  | f | 97.27 | 97.22 | 98.18 |
|  | g | 100.00 | 83.33 | 100.00 |
|  | h | 99.09 | 77.78 | 99.09 |
|  | i | 99.09 | 97.22 | 99.09 |
|  | j | 97.27 | 94.44 | 98.18 |

Table 9: Accuracy in the probing rounds per slot value.

```
TEMPLATE G.2.1
WHAT: Travel
FROM: Cologne
TO: Lisbon
BY: Overnight bus
CLASS: The most comfortable
WHEN: Anytime next week
```

```
TEMPLATE G.2.2
You are a customer of a travel agency. Here is
a description of the details of the travel you
want to make:

$INSTANCE$

The travel agent does not know about it yet.

Questions from the travel agent will start with
TRAVEL-AGENT. Please reply in the form: ANSWER:
<some text>

I will also ask you questions. These questions
will start with ME: .   Please answer these
questions with: ASIDE: <some text>

Important: Give short, direct answers!

Let us start.
```

```
TEMPLATE G.2.3
TRAVEL-AGENT: $AGENT-QUESTION%
```

```
TEMPLATE G.2.4
ANSWER: $ANSWER$
```

```
TEMPLATE G.2.5
ME: %CG-QUESTION%
```

```
TEMPLATE G.2.6
ASIDE: %CG-REPLY%
```

Figure 34: Travel agency version of the scorekeeping dialogue game. From top: Example Instance, Initial Prompt for Customer, Next-Round Template for Main Task, Response Parsing Schema for Customer Action, Next-Round Template for Probing Task, Response Parsing Schema for Reply to Probing Question.

```
TEMPLATE G.2.7
WHAT: Job Application
BACHELOR: Mechanical Engineering
INDUSTRY-EXPERIENCE: Two years
HIGHEST-EDUCATION: PostDoc
OTHER-SKILLS: Software development
AVAILABILITY: From September
```

```
TEMPLATE G.2.8
You are an applicant in a job interview. Here
is a description of your CV:

$INSTANCE$

The recruiter does not know about it yet.
Questions from the recruiter will start with
RECRUITER. Please reply in the form: ANSWER:
<some text>

I will also ask you questions. These questions
will start with ME: .  Please answer these
questions with: ASIDE: <some text>

Important: Give short, direct answers!

Let us start.
```

```
TEMPLATE G.2.9
RECRUITER: $AGENT-QUESTION%
```

```
TEMPLATE G.2.10
ANSWER: $ANSWER$
```

```
TEMPLATE G.2.11
ME: %CG-QUESTION%
```

```
TEMPLATE G.2.12
ASIDE: %CG-REPLY%
```

Figure 35: Job Interview version of the scorekeeping dialogue game. From top: Example Instance, Initial Prompt for Job Applicant, Next-Round Template for Main Task, Response Parsing Schema for Applicant Action, Next-Round Template for Probing Task, Response Parsing Schema for Reply to Probing Question.

```
TEMPLATE G.2.13
WHAT: Restaurant
APPETIZER: Fries
DRINK: Orange juice
MAIN-DISH: Burrito bowl
DESSERT: Strudel
SALAD: Avocado salad
```

```
TEMPLATE G.2.14
You are a customer in a restaurant. Here is a
description of what you'd like to order:

$INSTANCE$

The waiter does not know about it yet.

Questions from the waiter will start with
WAITER. Please reply in the form: ANSWER: <some
text>

I will also ask you questions. These questions
will start with ME: .  Please answer these
questions with: ASIDE: <some text>

Important: Give short, direct answers!

Let us start.
```

```
TEMPLATE G.2.15
WAITER: $AGENT-QUESTION%
```

```
TEMPLATE G.2.16
ANSWER: $ANSWER$
```

```
TEMPLATE G.2.17
ME: %CG-QUESTION%
```

```
TEMPLATE G.2.18
ASIDE: %CG-REPLY%
```

Figure 36: Restaurant version of the scorekeeping dialogue game. From top: Example Instance, Initial Prompt for Client, Next-Round Template for Main Task, Response Parsing Schema for Client Action, Next-Round Template for Probing Task, Response Parsing Schema for Reply to Probing Question.

```
TEMPLATE G.2.19
WHAT: Numbered letters
A: 3010
B: 2345
C: 6666
D: 7666
E: 4353
F: 6570
G: 5656
H: 9212
I: 2882
J: 7004
```

```
TEMPLATE G.2.20

Numbers have been assigned to letters, but the
questioner does not know about it yet. Here is
the mapping:

$INSTANCE$

Questions from the questioner will start with
QUESTIONER. Please reply in the form: ANSWER:
<some text>

I will also ask you questions. These questions
will start with ME: .  Please answer these
questions with: ASIDE: <some text>

Important: Give short, direct answers!

Let us start.
```

```
TEMPLATE G.2.21
QUESTIONER: $AGENT-QUESTION%
```

```
TEMPLATE G.2.22
ANSWER: $ANSWER$
```

```
TEMPLATE G.2.23
ME: %CG-QUESTION%
```

```
TEMPLATE G.2.24
ASIDE: %CG-REPLY%
```

Figure 37: Abstract version of the scorekeeping dialogue game using numbered letters. From top: Example Instance, Initial Prompt for Answerer, Next-Round Template for Main Task, Response Parsing Schema for Answerer Action, Next-Round Template for Probing Task, Response Parsing Schema for Reply to Probing Question.

```
TEMPLATE G.2.25
WHAT: Things at places
LEFT: piano
RIGHT: pencils
TOP: pens
BOTTOM: skirts
CENTER: bees
NORTHWEST: eyeliner
NORTHEAST: pants
SOUTHWEST: apples
SOUTHEAST: pineapple
HERE: napkin
THERE: camellia
NOWHERE: utility blade
EVERYWHERE: mascara
INSIDE: whale
OUTSIDE: rhodium
```

```
TEMPLATE G.2.26

Things are in different places, but the
questioner does not know about it yet. Here is
the situation:

$INSTANCE$

Questions from the questioner will start with
QUESTIONER. Please reply in the form: ANSWER:
<some text>

I will also ask you questions. These questions
will start with ME: .  Please answer these
questions with: ASIDE: <some text>

Important: Give short, direct answers!

Let us start.
```

```
TEMPLATE G.2.27
QUESTIONER: $AGENT-QUESTION%
```

```
TEMPLATE G.2.28
ANSWER: $ANSWER$
```

```
TEMPLATE G.2.29
ME: %CG-QUESTION%
```

```
TEMPLATE G.2.30
ASIDE: %CG-REPLY%
```

Figure 38: Abstract version of the scorekeeping dialogue game using things assigned to places. From top: Example Instance, Initial Prompt for Answerer, Next-Round Template for Main Task, Response Parsing Schema for Answerer Action, Next-Round Template for Probing Task, Response Parsing Schema for Reply to Probing Question.

| # | A | Game Master | B |
|---|---|---|---|
| 1 | | [A⟩GM] [initial prompt] | |
| 2 | | [GM⟩B] What is the next request? | |
| 3 | | | [GM⟨B] Where does your trip begin? |
| 4 | | [A⟨GM] TRAVEL-AGENT: Where does your trip begin? | |
| 5 | [A⟩GM] ANSWER: Cologne. | | |
| 6 | | [GM|GM] Cologne. | |
| 7 | | [GM|GM] Slot filled: True | |
| 8 | | [GM⟩B] Cologne. | |
| 9 | | [GM⟩B] What is the next request? | |
| 10 | | | [GM⟨B] Please let me know what kind of transport you want. |
| 11 | | [A⟨GM] TRAVEL-AGENT: Please let me know what kind of transport you want. | |
| 12 | [A⟩GM] ANSWER: Overnight bus. | | |
| 13 | | [GM|GM] Overnight bus. | |
| 14 | | [GM|GM] Slot filled: True | |
| 15 | | [GM⟩B] Overnight bus. | |
| 16 | | [GM⟩B] What is the next request? | |
| 17 | | | [GM⟨B] What kind of class do you wish? |
| 18 | | [A⟨GM] TRAVEL-AGENT: What kind of class do you wish? | |
| 19 | [A⟩GM] ANSWER: The most comfortable. | | |
| 20 | | [GM|GM] The most comfortable. | |
| 21 | | [GM|GM] Slot filled: True | |
| 22 | | [GM⟩B] The most comfortable. | |
| 23 | | [GM⟩B] What is the next request? | |
| 24 | | | [GM⟨B] When is the trip? |
| 25 | | [A⟨GM] TRAVEL-AGENT: When is the trip? | |
| 26 | [A⟩GM] ANSWER: Anytime next week. | | |
| 27 | | [GM|GM] Anytime next week. | |
| 28 | | [GM|GM] Slot filled: True | |
| 29 | | [GM⟩B] Anytime next week. | |
| 30 | | [GM⟩B] What is the next request? | |
| 31 | | | [GM⟨B] Where are you going to? |
| 32 | | [A⟨GM] TRAVEL-AGENT: Where are you going to? | |
| 33 | [A⟩GM] ANSWER: Lisbon. | | |
| 34 | | [GM|GM] Lisbon. | |
| 35 | | [GM|GM] Slot filled: True | |
| 36 | | [GM⟩B] Lisbon. | |

Figure 39: Scorekeeping: Excerpt of the slot filling turns for Claude.

| # | A | Game Master | B |
|---|---|---|---|
| 1 | | [GM|GM] Begin probing | |
| 2 | | [A⟨GM] ME: Is the travel agent aware of your origin? Please answer yes or no. | |
| 3 | [A⟩GM] ASIDE: Yes. | | |
| 4 | | [GM|GM] yes | |
| 5 | | [GM|GM] Answer for FROM valid after 1 tries. | |
| 6 | | [GM|GM] Answer is correct. | |
| 7 | | [A⟨GM] ME: Do you believe that the travel agent is aware of where you are going? Please answer yes or no. | |
| 8 | [A⟩GM] ASIDE: Yes. | | |
| 9 | | [GM|GM] yes | |
| 10 | | [GM|GM] Answer for TO valid after 1 tries. | |
| 11 | | [GM|GM] Answer is correct. | |
| 12 | | [A⟨GM] ME: Does the travel agent already know whether you go by plane or by train? Please answer yes or no. | |
| 13 | [A⟩GM] ASIDE: No. | | |
| 14 | | [GM|GM] no | |
| 15 | | [GM|GM] Answer for BY valid after 1 tries. | |
| 16 | | [GM|GM] Answer is incorrect. | |
| 17 | | [A⟨GM] ME: Is the travel agent aware of which class you would like to book? Please answer yes or no. | |
| 18 | [A⟩GM] ASIDE: No. | | |
| 19 | | [GM|GM] no | |
| 20 | | [GM|GM] Answer for CLASS valid after 1 tries. | |
| 21 | | [GM|GM] Answer is incorrect. | |
| 22 | | [A⟨GM] ME: Is the travel agent aware of the dates of your trip? Please answer yes or no. | |
| 23 | [A⟩GM] ASIDE: No. | | |
| 24 | | [GM|GM] no | |
| 25 | | [GM|GM] Answer for WHEN valid after 1 tries. | |
| 26 | | [GM|GM] Answer is incorrect. | |
| 27 | | [GM|GM] End probing | |

Figure 40: Scorekeeping: Probing at the last round for Claude.