# OpenReview forum: "clembench: Using Game Play to Evaluate Chat-Optimized Language Models as Conversational Agents"
_EMNLP/2023/Conference — EMNLP 2023 Main_

### Official Review · Reviewer_LsBC · 2023-07-31

**Soundness:** 3

**Excitement:**

3: Ambivalent: It has merits (e.g., it reports state-of-the-art results, the idea is nice), but there are key weaknesses (e.g., it describes incremental work), and it can significantly benefit from another round of revision. However, I won't object to accepting it if my co-reviewers champion it.

**Paper Topic And Main Contributions:**

This paper introduces a new suite of benchmarks for evaluating chat-oriented LLMs.
The benchmarks are designed as two-player games that can be played through a text interface and assess certain representational domains (e.g. world modeling).
The basic premise is that agents interact with the text games and must demonstrate situated language understanding (SLU) in order solve the specified task.
To mediate and assess the performance of the models, an automated game master which strictly checks responses for formatting and game rule violations is utilized.
The paper introduces five text-based games: the word game taboo, a wordle clone, a drawing game, a picture matching game, a knowledge exchange and probing task.
The framework also supports the addition of new text-based games by future researchers.

The paper then describes a set of experiments where a collection of chat-oriented LLMs such as Claude, Vicuna, GPT-3.5/4, etc are tasked with solving these text-based games.
Unfortunately, I saw no information regarding how the models were prompted: was it zero-shot or few-shot, how were the prompts constructed (there is no reference to this in the main body of the paper, but the appendix includes some details), sampling hyperparameters, etc.
The results do seem to indicate models that are known to be fairly strong, such as GPT-3.5/4 and Claude, outperform many of the open models such Open Assistant, with GPT-4 achieving the highest scores.

**Questions For The Authors:**

* Why do these need to be two-player games? Why transform single-player games into two-player? Is this purely for convenience or does it have some diagnostic intent?
* How are the mixed LLM games meant to be interpreted? How does agent/agent play assess ability? Imagine an agent excels at one role and not the other, won't it fail the test?
* You refer to the use of world models multiple times. Is this an assumption that LLMs build an implicit world model, or are you refering to something explicit like the use of an autoencoder or knowledge graph?
* How were the quality scores for each game determined? For example why use F1 for the drawing game, while using simple binary measures for the picture reference game?
* What is the methodology for running the experiments: is it zero-shot or few-shot, what were the sampling hyperparameters?

**Reasons To Accept:**

* The paper introduces a new framework for testing chat-oriented LLMs for situated language understanding with an associated set of benchmarks.
* The benchmark suite is open source and will be released upon acceptance and supports the addition of new tests in the future.
* The test sets for the benchmark suite have been created from scratch, so there is little risk of overlap with the training data of LLMs.

**Reasons To Reject:**

While this paper may hold promise, there are numerous issues that require a major rewrite to clearly explain why a new benchmark is needed and how it improves upon existing benchmarks:

* There is no comparison to any existing benchmarks to explain how or why this approach is better than existing evaluation approaches. It's not clear how this approach addresses any gaps in situated language understanding (SLU) that might be present in previous evaluation approaches?
* Why are two-player text-games a good proxy for assessing SLU? This is even more confusing because one of the games is inherently a single-player game and is transformed into a two-player game to fit into this framework. Furthermore, the experiments include examples where different models take on the role of Agent 1 and Agent 2. What research question is being answered by these mixed LLM benchmarks? (Note that the paper references Schlangen 2023 [1], a preprint from April, to motivate the approach, though that paper doesn't seem to address the need for multi-agent games, only that they are possible and likely different from LLM training.)
* There is no human validation for these benchmarks. Would these benchmarks align with human judgements (which is typically desired)? If that's unnecessary for some reason through the design of the benchmarks, then maybe that should be explicitly called out. Rather the only mention of human performance in the paper is the expectation that humans would perform well on the tasks (again without any validation, simply stating that the tasks are relatively easy).


### Updates After Rebuttal

While I still strongly believe the authors should provide more concrete validation for their claims, ideally in the form of a human evaluation, in discussion with the other reviewers, I've decided to raise my soundness score from 2 to 3. Since at least some of the rankings match intuition GPT-3 < GPT-3.5 < GPT-4, there is some credence that this approach might mirror human expectations. Since the authors intend this ranking to be general across current and future LLMs, I do not think the methodology can fully support their claims, which is why I do not provide a soundness score of 4.

[1] David Schlangen. 2023. Dialogue games for benchmarking language understanding: Motivation, taxonomy, strategy.

**Reproducibility:**

3: Could reproduce the results with some difficulty. The settings of parameters are underspecified or subjectively determined; the training/evaluation data are not widely available.

**Reviewer Confidence:**

4: Quite sure. I tried to check the important points carefully. It's unlikely, though conceivable, that I missed something that should affect my ratings.

**Typos Grammar Style And Presentation Improvements:**

* I think less time could have been spent on each individual game, and more on the basic premise to better frame why the approach makes sense and how it helps evaluation. Rather, things like how these games were encoded for LLMs seems more important (especially considering the expectation that future researchers can extend the set of games), while one specific game could be detailed in the main body with the remaining detailed with figures in the appendix. The main paper could focus on why these four games were chosen and how each advances situational language understanding in a way that cannot currently be done.
* Maybe these language games should be validated by humans. Even a small pilot study would be helpful. First, to see human performance (rather than assume humans can easily solve the tests) and second, to assess the relative performance of the various models on these questions in a more free-form way (one that does not require the strict checking of the game master) to have a way to determine an expected performance of the models and see how close the proposed framework aligns with that (usually new benchmarks are validated against human judgements).
* The figures are not clear and definitely need explanation. What do the different colors indicate, what is fed into the LLM at each step, etc.
* Figure 3 is not understandable at all. I'd honestly get rid of it and simply explain it using text (it would likely take up much less space too).
* Figure 2 needs more explanation regarding the terminology used. What is incremental processing in this framework (all LLMs assessed in this paper incrementally process one token at a time; if your conception of incremental processing is different, then you need to state that explicitly in the main body of the paper when you introduce the term).

---

> ### Author Rebuttal · Authors · 2023-08-28
>
> We thank reviewer LsBC for their very thorough engagement with our work, and hope that we can clear up some of the (very helpful and interesting!) issues raised.
>
> We will start with the direct questions and then address the issues raised elsewhere in the review.
>
> **Q1: Why two-player games?**
>
> It isn't necessarily the two-player nature that we're after (one could well imagine interesting games with *more* players) but rather the round-based interactivity. In that sense, wordle is easily framed as a two-player game, with one "player" -- the feedback-giver -- being fully rule-based. The overall effect is still one of creating a dialogue context for the guesser, in which earlier material (the earlier hints) needs to be integrated (what is the overall information that now constrains the guess).
>
> But in general, we're interested in covering the whole continuum, which is why we've included a single-round game (the reference game) as well.
>
>
> **Q2: How are mixed LLM games to be interpreted? How does agent/agent play assess ability? Imagine an agent excels at one role and not the other, won't it fail the test?**
>
> In the case where the same model-type is used for both player roles, we can assess, but not separate, the strength of capabilities needed for either role, and indeed in an overall sense the ability to be responsive to the other player's contribution. This, we contend, is a good basis for relative ranking of models.
>
> As an additional opportunity for finer-grained analysis, the mixing of models can bring out separate strengths: For the games that have asymmetric roles (e.g., taboo, with the hint giver and the guesser), are certain models better at one or the other role? (This has to be expanded on in future work, for which this paper is meant to lay the foundation.)
>
>
> **Q3: You refer to the use of world models multiple times. Is this an assumption that LLMs build an implicit world model, or are you referring to something explicit like the use of an autoencoder or knowledge graph?**
>
> Thank you for this question, which allows us to emphasise again an important point: We are aiming for a measurement of underlying capabilities, expressed and accessed only through behaviour. As such, the test instrument is agnostic as to how the capability is realised in the model. The claim here is that to successfully play some of the games, a certain capability likely needs to be present. How it is realised -- implicitly or more explicitly guided by the model designer -- is an independent question. Should the paper be accepted, we will stress this point further.
>
>
>
> **Q4: How were the quality scores for each game determined? For example why use F1 for the drawing game, while using simple binary measures for the picture reference game?**
>
> We aimed for a certain degree of fine-grainedness in each metric, so that where we can give partial credit (as in the drawing game), we do so. How to do so depends on the individual game (what can count as partial success). For the reference game, a more fine-grained measurement did not suggest itself (either the right object is picked or not, in a single-round game); and also, this follows precedent in the psycholinguistic literature that uses this game.
>
>
> **Q5: What is the methodology for running the experiments: is it zero-shot or few-shot, what were the sampling hyperparameters?**
>
> We will consolidate this information in a central place, should the paper be accepted. In short: The inference parameters are mentioned in Section 5 (temperature set to 0). The prompts are shown in the appendix. Many of them (besides the private/shared game) give examples of the expected form of the player moves, embedded in a free text; in that sense, they could be called few-shot. We arrived at these prompts through light experimentation with GPT-3.5 and other models. We need to stress again that prompt engineering is *not* our concern here -- we are after an instrument that can (fairly) distinguish, not after optimisation.
>
>
> **R1-a: Comparison with existing benchmarks**
>
> We are not claiming that this benchmark should *replace* existing reference- or preference-based ones. It is *complementary* to them in that it challenges interactive, context-dependent capabilities. Related attempts that directly target interactivity are only now starting to appear (in preprint form); we will use the additional space, should the paper be accepted, to acknowledge these.
>
> *R1-b: It's not clear how this approach addresses any gaps in situated language understanding (SLU) that might be present in previous evaluation approaches?*
>
> The standard benchmarks (GLUE, HELM, BIG-BENCH) are almost exclusively reference-based and static. On the other hand, most work on situated language understanding (e.g., TEACh, ALFRED, Minecraft) focusses on complex environments and uses purpose-built models. (It is not inconceivable that general foundation models will reach the required flexibility at some point. When they do, clemgame will be ready to provide the interface.)
>
>
> **R2: human validation**
>
> In the current state, the instrument needs to show its worth through its discriminative power between models (that is, the ability to rank them). This, we believe, it does.
>
> Having a human ceiling for performance is of course very desirable, but for now left to future work. Let us note that such a collection effort, if done correctly and for the number of games that we have here, requires quite a substantial effort --- which, if demanded for a single conference paper, would drive all but the highest funded industrial labs out of the game --; for now, we focussed on the already quite substantial effort required in laying the foundations. We believe that this does not take away anything from our argument as presented here.
>
>
>
> **reproducibility**
>
> We would like to point out that we not only promised to release the full code on publication, but we also made the effort of already releasing the code via an anonymous repository (URL in the paper), for full inspection. We hope that this alleviates worries about reproducibility.
>
>
>
> **holds promise [...] rewrite**
>
> We would like to ask LsBC to reconsider their assessment that the noted issues would "require a major rewrite". We believe that some of the issues have already been addressed in the paper (see above) and can easily be made more prominent in a final version, while we see some of the other issues as very valuable suggestions for further work, for which however the present paper needed to lay the (rather substantial) foundation.
>
> We read the substantial (and very valuable!) engagement that R3 has given this paper as a sign that they see productive value in the paper already (if only for generating further ideas!), and ask them to reflect this in their decision.

---

### Official Review · Reviewer_6HrB · 2023-08-07

**Soundness:** 4

**Excitement:**

4: Strong: This paper deepens the understanding of some phenomenon or lowers the barriers to an existing research direction.

**Paper Topic And Main Contributions:**

The authors present a new benchmark for evaluating LLMs using text-based games like wordle and taboo. They then conduct a thorough analysis of current SOTA language models (both open and closed source) and analyze their performance. They find that agents generally get better over time, but are not near human level performance, rendering this benchmark important for testing language understanding skills.

**Reasons To Accept:**

The idea behind evaluating language models using a game is clever, and provides a good benchmark for aspects of language understanding. The authors conduct a thorough analysis and provide useful results for the community.

**Reasons To Reject:**

None, I think the paper should be accepted.

**Reproducibility:**

5: Could easily reproduce the results.

**Reviewer Confidence:**

2: Willing to defend my evaluation, but it is fairly likely that I missed some details, didn't understand some central points, or can't be sure about the novelty of the work.

---

> ### Author Rebuttal · Authors · 2023-08-28
>
> We thank reviewer 6HrB for their very encouraging reception of our work and their appreciation of the underlying idea and its realisation.

---

### Official Review · Reviewer_dwAD · 2023-08-10

**Soundness:** 4

**Excitement:**

4: Strong: This paper deepens the understanding of some phenomenon or lowers the barriers to an existing research direction.

**Missing References:**

Although not absolutely necessary, this work can cite prior work (Aher 2022 https://openreview.net/pdf?id=eYlLlvzngu), which analyzes how large language models can imitate and simulate multiple humans. This can be paired with the Park 2023 work that they currently cite as works that simulate LLMs as agents in a multi-agent system from a HCI/sociological standpoint.

**Paper Topic And Main Contributions:**

This paper presents a benchmark to evaluate language models at interactive dialog-based games. The authors design games that chatbots can work through turn-by-turn. They test existing open-source and closed-source language models based on their abilities to complete each game successfully and on the quality of how the game was played (a metric designed specific to each game). They release their python benchmark as a GitHub repository and claim that community members can add new games and models from the community to their code.

Regarding the type of contribution they provide to EMNLP, the paper provides an NLP engineering experiment and open-source benchmark that can be used to evaluate different language models at their abilities to evaluate language models at their conversational agent capabilities.

**Questions For The Authors:**

A. Failure modes for GPT-4, GPT-3, and open-source models. It is mentioned in line 420-424 that GPT-4 does not reach 100% on percent played for all games. Can you provide specific examples within the drawing and taboo games where GPT-4 fails at this task, to motivate future work that models need to improve on?
B. Failure modes for Open-Source models - It is mentioned in the “Ethics Statement” that open source models are behind closed source models at the games provided in the clembench benchmark. Table 2 seems to support this to show that the open source models (ko/flc/ost/vcn) are all particularly bad at the non-wordle games. What do these failures look like in non-wordle games? Do there exist common themes in the types of mistakes that existing open-source models are making that closed-source models are doing comparatively better at?
C. New games - Given that GPT-4 completes most games with 100% success rate, what types of new games do the authors think that the authors/community members can design where GPT-4 wouldn't be able to successfully complete the game?

**Reasons To Accept:**

1. Novel benchmark - This paper provides a new benchmark to evaluate language models. It fills an existing need to test language models at a deeper level, particularly at "interactive dialogue aspects of chat-based LLMs". As such, it is valuable to the broader NLP community.
2. Analysis of existing language models - This paper demonstrates that newer, larger, and proprietary LLMs tend to be better at these simple dialogue games.

**Reasons To Reject:**

Insufficient qualitative analysis of failure modes - The paper makes claims that newer, larger models tend to be better at the games they introduce (83-85). As such, the paper provides experiments (as seen in Table 2) to substantiate these claims. However, to better understand why models fail these types of dialogue games, more qualitative analysis should be included as to what failure modes look like for both older, smaller models and newer, larger models. For instance, what do failing models tend to output for games that they fail at? Are there trends or common themes amongst these outputs?

**Reproducibility:**

5: Could easily reproduce the results.

**Reviewer Confidence:**

4: Quite sure. I tried to check the important points carefully. It's unlikely, though conceivable, that I missed something that should affect my ratings.

**Typos Grammar Style And Presentation Improvements:**

Sentence “While the public… not a clever dog” (34-45) can be split into 2-3 shorter sentences for conciseness and clarity.

---

> ### Author Rebuttal · Authors · 2023-08-28
>
> We thank reviewer dwAD for their receptive and encouraging comments.
>
> We would like to take the opportunity to address some of the issues raised and the questions.
>
> **Q-A & Q-B: more discussion of the failure modes for the otherwise successful models (GPT-3.5 & 4), and for the open source models.**
>
> This is an excellent suggestion. Should the paper be accepted, we will use the additional page to move some of the existing discussion of failure cases from the appendix into the main part. We will also provide more examples both on the instruction-following level (wrongly formatted, and hence non-parseable, output) and the game-playing level (wrong / unhelpful game moves).
>
> We will also stress more what, according to our findings, at the moment marks the greatest difference between locally-installable and closed models, namely the ability to follow instructions on the expected *form* of the output. Our hypothesis is that at the current state, the locally-installable models are insufficiently trained on instruction following to even reach the gameplaying stage reliably.
>
>
> **Q-C: saturated for GPT-4? new games?**
>
> The question of the predictive value of the instrument is a great one. We think that the work described here has laid the foundation for many more detailed investigations, which will have to show how predictive performance on one game is for another (and hence, to what extent our analysis of the degree of involvement of various aspects of the underlying capability is correct).
>
> When it comes to GPT-4's performance, however, we need to point out that there might be a slight misunderstanding here: While GPT-4 has high success in playing the games as such (that is, following instructions; as measured by % played), there is still ample room for improvement when it comes to successfully playing the games (as measured by the quality score). [You can think of this as the difference between being able to make legal moves in chess on the one hand, and winning the game on the other.] In that respect, we consider our benchmark not yet saturated, and there are clear areas for improvement even for GPT-4 (for example, in reasoning with and integrating over the whole interaction the letter-based feedback in wordle).
>
>
> Thank you for the reference to the Aher et al paper, which we will include, and for the suggestion for stylistic improvements, which we will implement, should the paper be accepted.

---

### Meta-Review · Area_Chair_JQbE · 2023-09-12

**Recommendation:** 4

**Metareview:**

Overall, this paper makes interesting contributions to evaluating LMs as agents by framing the evaluation paradigm as gameplay.

The paper addresses an important topic, takes an innovative (though debatable) approach, arrives at intriguing conclusions, and ultimately could inspire future work as the reviewers collectively describe. With that said, I do think there are important questions around design and decision-making with the games, largely echoing the comments of Reviewer LsBC in their thorough review. More broadly, as the paper partially describes, I am left dissatisfied with the current understanding of the validity/reliability of the benchmark (i.e. the core matters for assessing any measurement approach), the underlying theoretical understanding of the construct being evaluated (I think the paper does an alright job, deferring partially to prior work, but we really need a very good understanding of what we are trying to measure before we can say we are doing it well), and ultimately on how much this buys us over other evaluations. To me, the last point is the biggest question: its clear the evaluation regime here is very distinctive/different, but if it doesn't new/different insights, what's the point? I would like to see the authors engage with this as the core matter if the paper is accepted.

---

### Decision · Program_Chairs · 2023-10-07

**Decision:**

Accept-Main

**Comment:**

Overall, this paper makes interesting contributions to evaluating LMs as agents by framing the evaluation paradigm as gameplay.

The paper addresses an important topic, takes an innovative (though debatable) approach, arrives at intriguing conclusions, and ultimately could inspire future work as the reviewers collectively describe. With that said, I do think there are important questions around design and decision-making with the games, largely echoing the comments of Reviewer LsBC in their thorough review. More broadly, as the paper partially describes, I am left dissatisfied with the current understanding of the validity/reliability of the benchmark (i.e. the core matters for assessing any measurement approach), the underlying theoretical understanding of the construct being evaluated (I think the paper does an alright job, deferring partially to prior work, but we really need a very good understanding of what we are trying to measure before we can say we are doing it well), and ultimately on how much this buys us over other evaluations. To me, the last point is the biggest question: its clear the evaluation regime here is very distinctive/different, but if it doesn't new/different insights, what's the point? I would like to see the authors engage with this as the core matter if the paper is accepted.